# Thermodynamics in Ecology—An Introductory Review

**DOI:** 10.3390/e22080820

**Published:** 2020-07-27

**Authors:** Søren Nors Nielsen, Felix Müller, Joao Carlos Marques, Simone Bastianoni, Sven Erik Jørgensen

**Affiliations:** 1Department of Chemistry and Bioscience, Section for Sustainable Biotechnology, Aalborg University, A.C. Meyers Vænge 15, DK-2450 Copenhagen SV, Denmark; 2Department of Ecosystem Management, Institute for Natural Resource Conservation, Christian-Albrechts-Universität zu Kiel, Olshausenstrasse 75, D-24118 Kiel, Germany; fmueller@ecology.uni-kiel.de; 3MARE—Marine and Environmental Sciences Centre, Department of Life Sciences, University of Coimbra, 3000-456 Coimbra, Portugal; jcmimar@ci.uc.pt; 4Department of Earth, Environmental and Physical Sciences, University of Siena, Pian dei Mantellini 44, 53100 Siena, Italy; simone.bastianoni@unisi.it; 5Department of General Chemistry, Environmental Chemistry Section, Pharmaceutical Faculty, University of Copenhagen, Universitetsparken 2, DK-2100 Copenhagen Ø, Denmark

**Keywords:** energy, exergy, entropy, minimum dissipation, maximum entropy production, maximum exergy storage, far-from-equilibrium systems, thermodynamics of life, negentropy

## Abstract

How to predict the evolution of ecosystems is one of the numerous questions asked of ecologists by managers and politicians. To answer this we will need to give a scientific definition to concepts like sustainability, integrity, resilience and ecosystem health. This is not an easy task, as modern ecosystem theory exemplifies. Ecosystems show a high degree of complexity, based upon a high number of compartments, interactions and regulations. The last two decades have offered proposals for interpretation of ecosystems within a framework of thermodynamics. The entrance point of such an understanding of ecosystems was delivered more than 50 years ago through Schrödinger’s and Prigogine’s interpretations of living systems as “negentropy feeders” and “dissipative structures”, respectively. Combining these views from the far from equilibrium thermodynamics to traditional classical thermodynamics, and ecology is obviously not going to happen without problems. There seems little reason to doubt that far from equilibrium systems, such as organisms or ecosystems, also have to obey fundamental physical principles such as mass conservation, first and second law of thermodynamics. Both have been applied in ecology since the 1950s and lately the concepts of exergy and entropy have been introduced. Exergy has recently been proposed, from several directions, as a useful indicator of the state, structure and function of the ecosystem. The proposals take two main directions, one concerned with the exergy stored in the ecosystem, the other with the exergy degraded and entropy formation. The implementation of exergy in ecology has often been explained as a translation of the Darwinian principle of “survival of the fittest” into thermodynamics. The fittest ecosystem, being the one able to use and store fluxes of energy and materials in the most efficient manner. The major problem in the transfer to ecology is that thermodynamic properties can only be calculated and not measured. Most of the supportive evidence comes from aquatic ecosystems. Results show that natural and culturally induced changes in the ecosystems, are accompanied by a variations in exergy. In brief, ecological succession is followed by an increase of exergy. This paper aims to describe the state-of-the-art in implementation of thermodynamics into ecology. This includes a brief outline of the history and the derivation of the thermodynamic functions used today. Examples of applications and results achieved up to now are given, and the importance to management laid out. Some suggestions for essential future research agendas of issues that needs resolution are given.

## 1. Introduction

The enigmatic evolution of ecosystems as well as their phenomenology during development has for long been a puzzle to ecological researchers and has led to the current wish to improve our understanding of these our study objects [1,2,3,4,5,6,7,8,9,10,11]. This wish is a strong, if not the strongest, incentive to what we all do in this area of research and our curiosity is the driving force towards fulfilling the wish. We hope to find answers to questions, such as: what are the ecological backgrounds for special life cycle histories of individuals or populations? Or how do ecological societies reach a homeostasis or balance with their surroundings? Such questions will eventually lead to more fundamental ones, addressing the possibility to identify the ultimate causes of evolution for instance at the ecosystem level. What determines how the ecosystem or nature behaves and evolves as it does?

This type of questions has received an increasing attention from ecological researchers and has been addressed within the area of ecosystem theory by Müller and Leupelt [12]. To deal with ecology from a theoretical point of view is nothing new, but research has usually been carried out at species or population level, for instance May’s analyses of the relation between diversity and stability [13]. Raising these problematics to the ecosystem level does not only offer new challenges but determines also a considerable change in the character in the complexity of the problem and in the questions to be answered. Introduction of thermodynamics into ecosystem studies makes these aspects even more relevant [7,14,15,16]. 

As indicated in the formulations above, the questions may be raised at various levels of the ecological hierarchy. Thus, the attempts to answer them will span from the level of the individual or population through autecological, and dem-ecological studies, via syn-ecological studies of societies to whole ecosystems and in some cases even to the global, or biosphere level [17,18,19,20]. Each level might possess its own methods or strategies.

In addition to the hierarchical perspectives, the question may be addressed at different time and space scales, which poses fundamental, well known problems to ecologists, so as how, when and where to carry out sampling. The answers to the problem to be solved, regarding ecosystem behavior, may vary if applied to various time scales. What determines the annual cycles of for instance a wet meadow or a bog will differ from what determines the long-term behavior of the same system seen over a period of several decades or even centuries, not considering the potential changes induced by an increase in greenhouse effect. Referring to the spatial scale we may take the patchiness of plankton and macrophytes of aquatic ecosystems or terrestrial vegetation as examples of the complexity we face and have to explain.

The questions are difficult to answer alone due to the level of complexity we are dealing with. Meanwhile there are two “easy” ways of escaping this problem. First, one could take the attitude that the complexity of the problem is so immense that it is unsolvable and one should not pose that type of questions. Second, one could take a vitalism-oriented perspective and assume intrinsic or even divine powers governing the behavior. This attitude would leave us as passive elements outside the system with no possibility to intervene and only be able to observe the system. Neither of the two extreme attitudes seems attractive nor forwarding. The first choice does not leave much challenge, the second includes the answers to the question in itself.

On the other hand, the two “solutions” do not match with today’s generally materialist view of nature shared by most researchers. Again, here two different attitudes may be taken in order to solve the problem. The first, and traditional, represents the reductionist way of thinking. If only we have time, patience, and money enough we will eventually create a sufficient and adequate knowledge to understand the above problems fully. A second attitude would be to argue that reductionist science will never achieve the goal of answering such questions, and that consequently a material, holist approach must be taken. Although, the approaches presented in this paper tend to represent the latter direction, we will not aim to resolve the debate of reductionism vs. holism. This is considered to be too philosophical for the scope of this paper. Rather, we represent the pragmatic attitude that both approaches are needed, and that time, in addition to money, is the dominant constraint if ecologists are to solve the environmental problems we are facing today and in the future.

Returning to the point of ecosystem evolution in time, the idea that these dynamics are following a pattern or goal is not new in ecology. Several proposals to the driving forces of nature have been given throughout this century (for an overview of major issues in ecology see Table 1), maybe even since Darwin’s time if we accept his ideas about the role of selection in evolution as representing such a driving force.

Meanwhile, it took some years to implement these ideas to the level of ecology and ecosystems. First, ecological science needed to build up knowledge. Second, the systems approach, the introduction of general systems thinking [50,51] to ecology, was needed. An outline of historical events, within the science of physics and biology, as well as society, considered to be important to the application of thermodynamics in ecology is shown in Figure 1. Events are indicated by authors placed in diagram in accordance with indications of approximate time of events and area of contribution.

E.P. Odum’s 24 principles [2], proposed already in the 2nd edition of his *Fundamentals of Ecology* [2], seem to be one of the first attempts to describe a systematic and pattern-like behavior in the evolution of ecosystems through time. This is under the assumption that the ecosystem is not disturbed by outside forces, such as catastrophic events or human interference. The well-known principles deal with behavior at various levels addressing features such as the community energy and structure, life histories, nutrients cycling, selection pressure and homeostasis of the ecosystem. In spite of the powers and wide importance of these principles, the principles as a whole have been presented in surprisingly few other textbooks than Odum’s own. A recent demonstration of these powers may be found in Jørgensen et al. [52] and Nielsen et al. [53]. The weakness, if any, may be that the concepts are mainly phenomenological in character and not all parameters are easy to quantify. In other words, they really do not address the causality behind the phenomena described.

The idea to explain this seemingly systematic behavior of natural systems soon came around. Thus, E.P. Odum’s brother, H.T. Odum [22] suggested that ecosystem function worked so as to optimize their maximum (useful) power [3,54]—a principle derived from Lotka’s papers [24,55,56] at the beginning of the 20th century. Meanwhile, the maximum power principle seemed to have received relatively little attention compared to the later derived concept of eMergy (truncation of embodied energy) where the number of publications has increased during the latest years [26,57,58]. Furthermore, H.T. Odum might be best known for his contribution in the area of ecological modelling and founder of ecological engineering [59,60,61]. Other approaches were soon to follow which basically took two different directions as entrance point—a network oriented and a thermodynamic oriented direction.

The network direction of ecosystem analysis took its starting point in the economically founded input-output analysis as introduced by Leontief in the 30′ies [62] and later formalized in the 60′s [63]. This approach was transferred to ecology by Hannon [64,65] and Finn [66,67] in a series of papers in the 70s. Their works became the fundament of two other ecological researchers, B.C. Patten [4,11,46] and R.E. Ulanowicz [5,6], both working with a general understanding of the ecosystem from a network perspective. 

The other direction found its entrance point in a thermodynamic interpretation of biological systems. They are to be understood as “far from equilibrium” systems in the sense put forward by the direction of thermodynamic science founded by Onsager [68,69], Prigogine and co-workers [29,30,70,71,72,73,74,75] and Martyushev [42,43,76]. For popular treatments of these views please refer to Prigogine and Stengers [77] and Nicolis and Prigogine [78]. 

The approach has of course received a lot of criticism, since many researchers consider thermodynamics to be a science dealing with ideal gases only, and ecosystems are definitely not only gases and cannot be reduced to such a view only [16,79]. Meanwhile, we consider this a problem already inherent in the science of physics itself, which we will therefore not try to resolve here. Rather, in our opinion the thermodynamic laws must be obeyed by all biological systems, and thus also by ecosystems. The question is only how the thermodynamic balances are handled in a manner that at the same time allows the build-up of an organized and efficient structure, that at the same time does not violate the mandatory message of the second law, namely that entropy must always be positive (or zero). The thermodynamic constraints, furthermore, are so fundamental to the evolution and behavior of ecosystems that an understanding within this physical framework is needed if we want to go further in the area of understanding long term ecological dynamics. This standpoint seems to be shared by several authors throughout the recent decades. Thus, several attempts to analyze the process of the origin and evolution of life and living systems, in general from a thermodynamic point of view haven been carried out and is found in current literature [80,81,82,83,84,85,86,87,88,89,90,91,92,93,94].

From both the above directions the idea emerged that one should be able to tell something about the qualitative state and functioning of ecosystems. How is the ecosystem operating in general? How is it affected from outside? What are the consequences if we interfere through human-societal activities? These are the questions we are often faced with from politicians and managers. Traditionally, ecology is a quantitative science which has found difficulties in meeting this kind of questions [95], since the research on ecosystems does not come out with answers to the practical evaluations such as: good, reasonable, or bad. Indeed, a concept like (bio)-diversity seems to have found its way into the thinking of politicians, but diversity is only one parameter. And what does it indicate? The many expressions used do not necessarily tell the same story about the ecosystem, e.g., [96,97,98,99]. Although, there were some attempts to relate the concepts of biodiversity and diversity [100,101,102]. Think of the controversies about the possible connection between diversity and stability [13,103], to mention just one example. Meanwhile, the entropy expressions used in calculations of diversities for instances for populations or landscapes [104,105,106] may be considered as descriptive to the states of systems only. These spatial descriptors/indicators will therefore be omitted from the more functionalist approach to the concept of entropy taken here.

The ideas of measuring and indicating the quality of ecosystem state and function seem to have at least temporarily, culminated in the invention of new ecological “buzz-words” such as, *sustainability*, *resilience*, *integrity* and *ecosystem health*. All of these concepts have been dedicated to assist us in determining which direction societal development should take place in the future, in order to allow following generations and human society to persist, or even improve. At the same time, we need to remove the social discrepancies that exist between the highly industrialized, western societies and the developing world as stated also through the recent 17 Sustainable Development Goals (SDG’s). Combining the above approaches must be seen as an attempt to get closer to a scientific definition of the political concepts. The definition being scientific in the sense that it has a good materialistic foundation in the natural sciences of physics, chemistry and ecology [16]. In the future more intensive studies on regulatory mechanisms influence by second order cybernetics and semiotics are to be expected [107,108].

As seen from this introduction, there are no doubts that the search for this “Grail” of ecology, to understand ecosystems better, in order to deal better with the environmental problems that we are facing, needs to be a multi-disciplinary task. The complexity of the problems, especially when including also societal problems, e.g., concerning the nexus of energy, food and water supply, is so vast that it will not be possible for single persons to understand and even less to solve the problems all together and alone. Therefore, the approach taken in this review also illustrates that supportive scientific elements must be adopted from other areas of natural science although our basic platform—the existing, sound science of ecology—has already been formed.

The following text will begin with a brief introduction to the history of thermodynamics setting the milestones of observations important for the later application to biology and ecology (Section 2). The thermodynamic laws are described in the following Section 3 in a qualitative and quantitative manner, but formulas/equation have been kept to a minimum. Readers that are either already familiar with these traditional views or simply want to skip the equations may jump to the section of fundamental concepts (Section 4) where clarification of the terminology used throughout this text is made. 

Hereafter, the extension of the thermodynamic laws to far from equilibrium conditions is described (Section 5). This includes an introduction to the Prigoginean world views of living systems as dissipative structures, (Section 5.1) [109] that move toward a state of minimum dissipation (Section 5.2) and evolve through instabilities and bifurcations (Section 5.3). By the introduction of the hypothesis of the *minimum dissipation principle* we are moving in the direction of extremal principles to biological systems [110,111,112,113,114,115]. Later this principle has been criticized [116,117] and additional proposal describing ecosystems tendencies to obey a principle of maximum entropy production has been introduced [42]. The extension has led to several candidates for new laws or principles within thermodynamics. The question is whether we are dealing with a new ecological law of thermodynamics (Section 5.4)

The concept of entropy have been used in studies of organisms and ecosystems through the establishment of entropy balances of the systems (Section 7.1). The proposal that biological systems including ecosystem should evolve in a way that they optimize their thermodynamic efficiency, i.e., maintaining largest structure at lowest price, expressed as exergy, is derived and explained (Section 7.2 and Section 7.3). Results from applications gained hitherto by using this way of analyzing ecosystem is presented through some examples (Section 7.4, Section 7.5 and Section 7.6) and compared (Section 7.7). Finally, a discussion of the problems met during this work is made and a direction for future work in the area together with its potentials in monitoring and evaluation of nature’s function is proposed (Section 8 and Section 9).

## 2. History

The science of thermodynamics is probably one of the most difficult areas of physics to access for layman, although even children may understand its messages at an intuitive level. It has had a relatively long history of development, involving many of the most remarkable scientists of physics over a long time span, and the area is still subject to development and discussion. An outline of the history is given in Table 2. 

In particular the extension of thermodynamics to living systems and even ecosystems has recently caused some controversies. The very beginning of the science dates back to the last century with the works of Sadi Carnot on the efficiency of steam engines in 1924 [119]. In the middle of the century, the two fundamental laws we will be dealing with were formulated by Clausius around 1865 [120]. But the area also found other contributors like Lord Kelvin [128]. Late in the century, the connection to statistical mechanics were laid out by Boltzmann in the late 19th century [121] and Gibbs, 1878 [122] opening up for other statistical interpretations, known as thermo-statistics, e.g., [129,130]. During much of the time there has been a continuous discussion on the relations between entropy, its descriptive role in analyzing distributions and its adjacent concepts of order and disorder [131,132,133,134,135,136]. An additional issue raises from many of these works namely that the isomorphism observed between e.g., statistical mechanics and diversity as stated by Rodrigues et al. [137], do not necessary allow us to conclude that we also deal with homeomorphism c.f. Nielsen [16].

Although early authors like Lotka [24,55,56] were aware that energy and competition for this resource play a fundamental role to living systems, it took some more years before a fundamental dilemma was solved. The fact that living, ordered systems were able to exist and even grow, based on irreversible processes that continuously lead to an increase in entropy and disorder, seemed to be self-contradictory, constituting a dilemma. The puzzle was solved by the argument that something else had to be involved—something able to decrease the entropy of the system. Logically this “something” needed to have a negative value and was therefore referred to as negentropy [123]. A better view may in fact be that the entropy produced is exported to the environment, thereby keeping the net balance negative—at least locally, while processes going on in the system at the same time leads to a gross positive entropy production [138,139,140].

A final platform for the understanding of the development and growth of living systems was put forward by Prigogine and co-workers, e.g., [29,74] understanding these systems as dissipative structures—far from equilibrium. This was a fundamental cut with the traditional science since it suddenly became legitimate to treat the systems as thermodynamic structures although they indeed existed under conditions far, far from thermodynamic equilibrium—conditions much further away than the state of ideal gases normally dealt with within classical thermodynamics. The approach also stated that these structures would develop in a certain direction, towards a minimum dissipation state. Another set of controversies—in short minimization vs. maximization of entropy formation—arose as the activity always at the same time results in increased dissipation, i.e., more entropy to be formed [141,142,143,144,145]. This apparent controversy still awaits further discussions and resolution, in particular at the area of ecology and ecosystems. 

Other approaches, like information theory [146], have come up claiming to be thermodynamic in their approach. Unfortunately, the use of the entropy concept within this area seems to have caused more confusion and contributing to more conflicts than it has actually solved. One major example may be found in the proposals set forward by Brooks and Wiley [82]. Therefore, the use of these approaches will not be presented here, and warnings will only be given when confusions have been made. This area for sure will need more elaboration and specification in the future. For attempts trying to combine the two directions the readers are kindly referred to [87,107,147,148,149].

In fact, a whole area with possibilities of confusing concepts and relations between them exists. This area deals with the relation between not only entropy and information [146,150,151,152], but also the possible relations with measures such as *order* and *complexity*. The relation between entropy and order is almost classical. Entropy is often understood and explained as *disorder* although this might be considered a misconception [117] as order/disorder often are used as intuitive, vaguely defined, non-quantifiable concepts Meanwhile, following the idea above, as a consequence, intuitively order must be connected to its opposite—negentropy [139,140]. Therefore, concepts like *organization* and *complexity* are preferred nowadays but they are vague terms too. The relation to organization seems in part to share destiny with the relation to order as this concept is also lacking a concise definition. We may consider the relation between entropy and complexity to have been sorted out through the works of Chaitin [153,154] on *algorithmic information complexity* (AIC). More work on the clarification and specification of the relations between the mentioned concepts will be needed in the future. For some papers aiming at resolving this debate, see Morowitz [155], Papentin [156,157], Hinegardner and Engelberg [158] and Stonier [152].

## 3. The Thermodynamic Laws

The following text is not going to be a compendium in thermodynamics. Only this section serves to give a short introduction to the perspectives considered to be relevant to the science of ecology. For deeper knowledge in this area we shall refer to the following [118,159,160,161,162,163,164,165,166,167,168,169,170] and the theses’ of Evans [171], Wall [172] and Kay [173], respectively. For introductions to the importance for biology and ecology as reflected in this paper, please refer to Allen [174], Ebeling et al. [175,176], Garby and Larsen [177], Morowitz [118,166], Müller and Nielsen [178] and Jørgensen [8].

The two thermodynamic laws essential to living systems and thus ecology are the first and the second. To repeat shortly, the first law deals with the constancy of energy and the second with the continuous increase of entropy by all real processes. Luckily, the numbering of these two laws is always the same. As if the area of thermodynamics was not confusing enough in itself, textbooks seem to present a varying number of laws, usually three to four, with a varying numbering, e.g., the third law, sometimes numbered as zero. Unfortunately, there is no way to avoid this confusion as it is already there.

Thermodynamics is an area of the science of physics that is probably one of the hardest to access and comprehend. This presentation will relate as much as possible to the biological relations of the area, and thereby differs from the normal presentations of physicochemical systems given in textbooks on the topic. The basic equations of this area meanwhile, we do owe to the normal presentations, although these have often been derived on system exchanging only heat and work, i.e., material flows are neglected.

The quite recent introduction in textbooks of measures of energetic efficiency, like exergy and availability has of course inspired much of the work done on ecological systems, e.g., Ahern [179], Brzustowski and Golem [180].

### 3.1. The First Law of Thermodynamics

The first law, as stated above, tells us that the energy of the universe is constant. Energy may never be created or destroyed. Meanwhile, energy may be of different forms, the most common examples as given in the scientific formulation later, are heat and work. Most readers will be familiar with other forms such as radiation, electrical and chemical energies. Although, the form of the energy may change we will thus always be able to track it, see for instance the section on Brillouin later.

The implementation of the first law into ecology has been much straight forward. It is the first law we apply to biological systems when we estimate energy budgets of animals, like in many (eco-) physiological studies. And it is the same law we use when making energy budgets of ecosystems, like the ones we see in the studies of e.g., E.P. and H.T. Odum [3,21,59]. Two illustrations of this principle applied to biology may be found in Figure 2a,b, showing the energetic balances of physiological processes and ecosystems, respectively.

The first law—in its simple(st) version—takes the following form:(1)ΔU=Q+W
which tells us that the internal energy, *U*, of a system, may change as a consequence of heat (*Q*) or work (*W*) added to, or delivered by the system. As a consequence the signs of *Q* and *W* may change. This form is considered to be the “scientific” and most consistent form, although other forms exist—where the common interest in having the systems to do work has resulted in the sign of *W* to be negative—adding up to the possible confusion [168] (this equation is often given as Δ*U* = *Q* − *W* where a directionality is already included in the equation, as work is engineering is what one want to get out of the system. This seems to be the predominant form seen in America, whereas the form used in the text is mainly used in Europe). The examples to illustrate this in traditional textbooks on thermodynamics are calculations on work done by pistons. It could be noted that a possible area of interest in energy analysis of any system should be focusing on the duality or rather complementarity between *Q* and *W*, which arises from this simple equation of balance. Energy will be either work or heat.

It is clear that the equation, in its above form, will have a limited importance to ecology. This is due to the fact that other, equally, or maybe even more dominant energy forms exist in biological systems, like for instance the chemical energy delivered with nutrients and food. The importance of this part of the energy will be introduced later. Usually heat, as input, is playing only a small role to certain animals such as exotherms, rather heat is important as output or respiration. Work done on a biological system is likewise difficult to exemplify although existing, e.g., the work done on aquatic organisms when they are passively moved around in the water column. Normally it is the work done by the living organisms that is of importance, the energy invested in the flight of migratory birds, or in hunting by predators. Shifting to the level of ecology makes it more difficult to find examples since organisms and populations usually are in thermal balances with their environment, i.e., following the temperature of their surroundings, and we hardly see them as doing any work, although in a physical sense they are.

Not surprisingly, the above formulation has found its widest application in the organismic oriented part of biological sciences, like physiology. Just think of physiological equations like the following:(2)Ingestion=Production+Respiration+Defecation+Excretion
in principle the equation explains what the energy of the food ingested by an organism is used for (compare also Figure 2a). In fact, such an equation formed the background of Lotka’s maximum power principle which in turn appears to have inspired H.T. Odum’s work and the maximum exergy storage principle of Jørgensen and co-workers.

The most important message to ecology is that we can calculate energy budgets for our systems in the same way as for the organisms above. But in moving to the ecosystem level other processes become important and dominant to the behavior of the system, e.g., Nielsen [16]. Those processes are the transfers of matter via the food chain or rather food network. The importance of this approach was initiated by the work of Lindeman [181], but was especially strengthened for ecosystems through the works of E.P. and H.T. Odum [1,2,54,59].

Measuring the energy storages and fluxes of the ecosystem allows one to get an overall picture of the ecosystem function. An example may be found in Figure 2b where the energy in a marine bay ecosystem is mapped according to Odum [2]. The storages and flows caused by solar radiation are mapped and sizes are indicated. The energy flows are in principle following two routes, either through the grazing/predation food chain or degradation/recirculation through the benthic, detrital feeding organisms in the sediments.

### 3.2. The Second Law of Thermodynamics

The second law finds its roots in the middle of the 19th century with the works of Carnot around 1824 [119] and Clausius in the 1860s [120]. It was Carnot who discovered the incomplete conversion of heat into work. But it took additional years to coin the term: Entropy, which we owe to Clausius. He also formulated the two basic thermodynamic laws, the first and the second together around 1865–68 [120]. For a collection of some essential historical papers giving an outline of the development of the second law we suggest that the readers refer to Kestin [182].

The second law tells us that, while energy remains constant in quantity something else is changed: energy is transformed and consequently its quality is changed (see about Brillouin below). This change in quality occurs in one direction only, resulting in a part of the energy which cannot be used for work any longer. This part of energy is said to be dissipated and eventually leads to the formation of entropy, *S* (equal to *Q*/*T*, *Q* being the previous heat exchange and *T* the absolute temperature, i.e., in Kelvin). As a result of the one-directional change, (global) entropy change is always positive (or non-negative), leading to the following formulation:(3)$$dS=\frac{dQrev}{T}\geq 0\;\;\;$$
which is a simple statement relating the entropy to the heat changes of the system. It should be noted that the value of 0 (zero) only is reached for reversible processes or at true thermodynamic equilibrium.

As stated earlier, our understanding of the concept of entropy is connected with “disorder” although a precise meaning of this concept is rarely given. So, the above equation tells us that (nearly) all processes lead to an increase in disorder. This may be caused by the connection to statistical mechanics to the expressions known as Boltzmann’s equation [121]: (4)$$S=k\ln W=-k\ln\frac{1}{W}=-k\ln p$$
where *k* is Boltzmann’s constant, *W* is number of possible microstates, and p is the probability of each of the microstates. For a number of distinguishable particles, with varying probabilities (non-equiprobable distribution) the Boltzmann-Gibbs equation [183]:(5)$$S=k\;\underset{i}{\sum }p_{i}\ln p_{i}\;\;\;\;\;\;$$
is used, where *p_i_* are the probabilities of different possible, distinguishable elements. Under conditions close to equilibrium, systems, like ideal gases, will move to a distribution of particles having the highest probability or highest entropy. This may in a slightly oversimplified version be illustrated by Figure 3.

These formulations relate entropy to a statistical arrangement of the parts of a system. Hence, entropy becomes related to a more or less “probable” arrangement of the parts. This connection intuitively seems to be very convenient for our understanding of biological systems. As previously remarked, e.g., Berry [81], however, no clear connection between the concept of order and the structure and organization of biological systems exist. Furthermore, by these relations to order and organization, the entropy concept becomes loosened from the stringency in its original strictly thermodynamic sense. For further discussion of these matters see Berry [81], Tiezzi [94] and several works of Stonier [151,152].

Staying close to classical thermodynamics and the classical potentials, the fundamental law may be formulated as:(6)dU=T dS−P dV
which implicitly tells us that the internal energy, *U*, is a function of *S* and *V* (*U* changes (*dU*) as either *S* or Volume, *V*, is changed).

Meanwhile, this equation does not include the contribution of most importance to biological and ecological systems, the contribution from material fluxes, i.e., chemical compounds, atoms or molecules, entering the system. Thus, we may formulate an even more general form:(7)$$dU=T\;dS-P\;dV+\underset{i}{\sum }\mu_{i}\;dn_{i}\;\;$$
where *μ_i_* is the molar chemical potential and *n_i_* the moles of type/element *i*, respectively. This form seems to be most relevant to biological systems. The importance to ecological systems will be described later.

The above equation also leads to Gibbs free energy, that is another energy fraction, important to biological systems. It is defined as, subtraction of the product of the independent variables multiplied by their partial derivatives of the function *U*, respectively, from the internal energy, *U* (for derivation see [168], thus:(8)G=U−TS+PV

Taking the derivative of this equation and inserting the above results of *dU* gives:(9)dG=dU−d(TS)+d(PV) 
which is the starting point for derivation of exergy (see later, Section 6 A,B).

Again for the open system we need to add a contributions from the other energies, e.g., the part belonging to mechanical works—such as kinetic and potential energy, and contributions from chemical processes in the system as well, which we shall see are of great importance to biological systems. In still other cases electro-chemical contributions may have to be included.

## 4. Some Fundamental Concepts

Some concepts in the thermodynamic terminology are fundamental to the understanding and establishing of thermodynamic balances for biological systems. They therefore need to be introduced to shape the understanding of how biological and ecological systems might be viewed as special domains of thermodynamics. For the sake of clarification and specification they will shortly be dealt with here. The fundamental topics to deal with—and to clarify—here are: (a)the various *types* of systems and(b)the relations between energy *form* and *quality*

As the terminology in some cases differs between textbooks, we will briefly introduce the terminology stressing the way it is used throughout this text.

### 4.1. Types of Systems

In thermodynamics, systems are in general divided into three different types, *isolated, closed* or *open* systems, distinguished by their varying permeability of their boundaries to either energy and/or matter.

#### 4.1.1. Isolated Systems—Or Adiabatic Systems

These (e.g., Katchalsky and Curran [184]) are systems which as the term says are totally isolated from the surroundings. This means that they receive or exchange no fluxes of neither energy nor matter to or from the systems. Hence, their evolution—according to the second law of thermodynamics—can take place in one direction only, towards increasing entropy, i.e., towards the state of highest probability and degree of randomness. This type of systems may be illustrated by Figure 4, Figure 5 and Figure 6.

#### 4.1.2. Closed Systems

Closed systems—the second fundamental type—are systems which are open to energy fluxes only, i.e., they are not open to fluxes of matter. The system boundaries are often referred to as *diathermal walls* indicating the possibility of energy exchange as heat, e.g., Katchalsky and Curran [184]. It should be noted that a flux of energy (e.g., heat) potentially may serve to organize matter already enclosed in the system as is the case of Bénard-cells. Engineering or work systems, that are systems exchanging energy as heat or work only are typical examples of this types of systems dealt with in textbooks. This type of systems may be illustrated as Figure 5.

#### 4.1.3. Open Systems

These are systems that are open to *both* energy and matter fluxes. Living systems are typical examples of this. The energy flows in these (biological) system are “used” to distribute and organize matter into structure by processes of self-organization, e.g., Popovic [185]. Energy loss through dissipation as mentioned above is unavoidable due to *irreversibility* of processes. Material losses are unavoidable too but are supposed to strive at a minimum, to a level dictated by necessity or forcing functions. The difference from the other systems may be illustrated by Figure 6.

Clearly, biological systems, ecosystems and in fact most systems we meet in our everyday life—are open and thus belong to the third type, which makes this the most interesting to us. Plants use energy from the sun and nutrients (matter) from the soils. Animals get their energy in material form (chemically bound energy) only, by for instance grazing or predation. Both plants and animals, loose energy through respiratory processes, evapotranspiration, respiration and transpiration, respectively. The biosphere is eventually also an open system, but we may consider the material flux from the space to be so small that we may reduce our understanding of the planet Earth to be a quasi-closed system [84,85,186,187,188,189].

### 4.2. Energy: Form and Quality

As mentioned above, while energy remains constant, its form can change. But, not only does the form changes so does the quality or the “value” of the energy. By value we here mean its ability to do work. A view like this allows us to compare and evaluate different forms of energy between one and another. According to Brillouin [127] (see Figure 7) the forms of energy having the highest value form—the highest capacity to do work—are for instance radiation and electronic energy. These high quality forms are often referred to as low entropy forms of energy. The form with the smallest energetic quality or “value” is heat which may be characterized as the (almost) final state of the energy degradation, and is thus considered a high entropy form. Heat may only do work by means of a temperature difference, i.e., heat flowing from high to low temperature reservoirs. As energy is transformed, the change of energy value (as defined previously) takes place in only one direction, from higher to lower value. Chemical energies, i.e., energy bound in the molecules of chemical compounds, are examples of intermediate forms. 

Energy in high quality form enters biological systems in a relatively few cases. The photo-pigments of autotrophic organisms are able to capture the high-quality energy from the sun during the process of photosynthesis. Eventually, the energy captured in this manner by the ecosystem, becomes the ultimate input and constraint of what is going on in any ecosystem. So, in a thermodynamic sense, ecosystems are all bottom-up regulated as argued by Nielsen [190]. Recent research also argues that recycling of matter through the detritus-bacterial link, which may also be considered belonging to a bottom-up regulatory mechanism, may exert a similar stabilizing effect on the system as a whole [191].

In a few other cases, solar radiation constitutes an important input (information/signal) such as when it is used in our visual systems [192]. Intermediate and low-quality forms, i.e., chemical bound energy and heat are—with the exception of autotrophic organisms - dominating in biological systems. Meanwhile, in general the energy value—and therefore also the importance of the latter—is much lower than that of the chemically bound energy.

The principle of energy transformation in biological processes may be illustrated by Figure 8. High quality energy is captured by autotrophic processes, like photosynthesis, and partly used for building up complex molecules. In this way high quality energy is transformed to chemical energy. Complex molecules may in turn be broken down, releasing energy and heat. The energy released is temporarily stored in chemical bindings, e.g., in ATP. Several of these energies may be add up and be used for chemical synthesis of new complex molecules or compounds, always with a heat production, i.e., dissipation of energy as a result. These observations lead to the concept of dissipative structures presented in the following.

## 5. Far from Equilibrium Thermodynamics

Building on and extending the works of Onsager [68,69,193], Prigogine and co-workers [29,30,124], built up a framework serving the purpose of understanding the function and existence of structures far from equilibrium such as biological ones. Here, it is worth to note that biological systems are much further away from equilibrium than the systems used for development of the theory. The works explain how structures, referred to as *dissipative structures*, may exist far from equilibrium even though dominated by irreversible, entropy creating processes. These ideas can be understood as an expansion and connection to Schrödinger’s thoughts about the importance of negentropy in the realization of life through the *order from disorder principle* [123]. 

Schrödinger wondered about the fact that living system were ordered systems. They had therefore to be able to exist in the spite of the second law which dictates the development of systems towards states of more disorder. His solution for this problem was that “something” had to counteract this fundamental principle thereby, thus circumventing the second law. That “something” should therefore be able to change the direction of evolution and would thus have to be a negative as opposed to the normal entropy, i.e., negative entropy, and was logically termed—*negentropy*. His observation lead to the famous formulation that living systems were feeding on negentropy, which laid out the foundation of his so-called *order from disorder principle* mentioned above.

Although, the term negentropy on one hand seems convenient to use and to facilitate the understanding of the problem, it may actually be rather unfortunate. Since we have already learned that entropy must always be positive or non-zero such a thing like negative entropy should not exist. Therefore, the statement should rather have been that living systems exploit an energy flow or gradient into infinity, i.e., exploiting the exergy as effectively as possible, which allows them to move to ever increasingly ordered states as compared with the surroundings. Open systems, like living systems, may be viewed as systems deviating (strongly) from thermodynamic equilibrium. They are organized and structured by sorting out the molecules of life processes, mainly dominated by the atoms C, H, N, O, P and S as indicated by Morowitz [166] in increasingly complex patterns. This principle is illustrated by Figure 9. Molecules may be sorted in space but also concentrated in special spatial patterns. The mechanisms will be explored further in the following.

### 5.1. Dissipative Structures

The entropy balance, *dS*, of a closed or open system, may according to Prigogine be described as:(10)$$dS=d_{i}S+d_{e}S$$
where *d_i_S* is the entropy change caused by internal processes, while *d_e_S* is caused by external exchanges. Whereas *d_i_S* is always positive as dictated by the second law, the second term of the equation, *d_e_S*, may be negative and numerically larger than *d_i_S*, which allows the resulting entropy balance of the system also to be negative. 

This balance may be illustrated by Figure 10. It should be noted that we here consider entirely irreversible processes which are typical in Nature. To illustrate the importance to life we consider the following possible outcomes where *dS* may be negative, zero or positive. In the negative case:(11)$$d_{i}S+d_{e}S<0$$
meaning that:(12)$$d_{e}S<-d_{i}S$$
or the equivalent statement:(13)$$d_{i}S<-d_{e}S\;\;\;\;$$
which is an ultimate thermodynamic condition or constraint to be met in order for life to exist. Looking at for instance Equation (12) this means that *d_e_S* has to be even less than the negative of the internal entropy production. Under these conditions living structures may occur and even grow. In the case of equality, a thermodynamic balance or homeostasis exists.

If the equation is positive, i.e.,
(14)$$d_{e}S>-d_{i}S\;$$
meaning that the entropy flow to the outside, *d_e_S*, does not compensate for the entropy created by internal processes, the living structures or system under consideration may exist only until internal resources have been used up, i.e., over a limited time span. Otherwise, this thermodynamic condition eventually means death. Various examples of life strategies to cope with such temporal imbalances can be found throughout biology among both plants and animals, e.g., hibernation of seeds, hibernation in bears.

In the end, this representation of the thermodynamic balances of system in terms of entropy seems incomplete as it for instances introduces the possible existence of negative entropies. As both balances are necessarily positive, we need to add to the picture that both the two entropies must be compensated for by imports of (high quality) energy in order for the system to persist.

### 5.2. Minimum Specific Dissipation

Meanwhile, during the reading of various authors dealing with entropy balances, it is sometimes unclear, which type of entropy production they are speaking about. Often the term “specific entropy production” is used, which is the balance equation we are interested in for biological systems [194,195]. In this case the correct formulation of the above equation should look like:(15)$$\frac{1}{m}\;\frac{dS}{dt}=\frac{1}{m}\;\frac{d_{i\;}S}{dt}+\frac{1}{m}\;\frac{d_{e}S}{dt}$$
where *m* is the mass of the system [195]. According to these authors *d_i_S* and *d_e_S* may be designated as the *entropy production* and the *entropy flow term*, respectively. Failure to distinguish between the two formulations (i.e., entropy production vs, entropy production density) may be the cause of many of the ongoing discussions and controversies between the various ways of applying thermodynamics to ecosystems. This is valid in particular when considering arguments around maximization and minimization of entropy productions.

The principle can be illustrated by Figure 11. Here, entropy production increases in the initial phases of evolution of the system through time as a structure is building up. At later stages of development, the entropy production of the system will reach a stable level and approach a state of minimum specific entropy production. 

### 5.3. Evolution through Instabilities

To complete this presentation of thermodynamic laws, two other new candidates have been proposed in the current literature as indicated in the previous section. First, the work of Prigogine and co-workers has led to statements that a continuous minimization of entropy production will eventually lead to instabilities of the systems [29,124]. 

The occurrence of such instabilities is considered to be the mechanism through which evolution of the systems takes place. This has sometimes been nominated as a fourth law of thermodynamics. This evolutionary principle may be illustrated by Figure 12 and Figure 13, which basically represents a symmetry breaking of the system. 

The evolution finds parallels in both the concepts of Gould’s punctuated equilibria [196] as well as Kaufmann’s order on the edge of chaos [197].

### 5.4. An Ecological Law of Thermodynamics?

We have already presented, in the introductory chapters, a number of concepts proposed as functions able to give information about the quality state of ecosystems. Such functions have often been referred to as *ecological orientors* or *indicators*. Some of the concepts have even been proposed to serve as goal functions in ecosystem development [5,6,7,12,44,47,198,199,200,201,202,203]. For a comprehensive review the reader is referred to Nielsen and Jørgensen [204].

Goal functions do have a semantic bias, leading the thoughts towards ideas of a teleological behavior of nature. This is not the case here. The term comes from applied mathematics where it has a strict sense of meaning, for instance as a function acting as attractor of a set of equations. As such, it has entered the world of ecological modelling.

Through the discipline of ecological modelling and ecosystem theory we have learned a lot about ecosystems, about their complexity, and how the system reacts as a whole when subsystems are coupled together. This has allowed us to establish reasonably adequate, precise, quantitative models in the sense that models demonstrate quantitative values between modelled compartments which are consistent with values observed in nature.

Meanwhile, one major lesson learned, has been, that models often fail, i.e., loose their predictive value as qualitative changes become involved. Qualitative changes may be exhibited as sudden shifts in species composition, like during succession, or maybe even changes in the whole ecosystem structure, as observed when ecosystems collapse as a consequence of pollution. Thus, the changes are in general induced by some fluctuations, sometimes intrinsic, i.e., due to internal reasons, but most often causes are found on the outside to the system, natural as well as humanly induced, as illustrated by Figure 14.

This has led to a new area of ecological modelling with the purpose of improving existing models by integrating the possibility of changing internal structures in the model during a simulation. This approach is often referred to as *structural dynamic modelling*.

That ecosystems react to changes in the prevailing conditions of the forcing functions that drive and affect them, like solar radiation, temperature or pollution, is nothing new to ecology. We are all too well aware of the responses of the organisms and the systems’ capabilities to such changes,—a process we usually refer to as adaptation. While this phenomenon is well known in a qualitative sense it is much more difficult to understand it in a quantitative manner as it will be demonstrated in the following.

Meanwhile, in addition to this, the idea has come around that some adaptational strategies exhibited by organisms or even ecosystems would be better than others, an understanding which is implicitly found in the concept of *fitness*. Therefore, such evolutionary strategies would be favoured, i.e., selected for in the evolutionary process.

Quite logically, it has been proposed that thermodynamic efficiency would be a parameter important to biological systems, and that this would be a property to be selected for at both organism level as well as ecosystem level. Thermodynamic efficiency may, as seen in the following be expressed as the exergy of the system. The whole idea may be viewed and understood as a translation of the concept of survival of the fittest into thermodynamics. The structure—being it an organism, population, bio-coenosis, or ecosystem levels—which is most fitted is the one performing with the highest thermodynamic efficiency, exploiting the imposed gradients in an optimal manner, thus resulting in the highest exergy. Introducing this view in the theory of evolution sets up a physicochemical framework for discussion of the (neo-)Darwinian theory of evolution thereby removing the foundation of any accusations of a tautological argumentation. This principle of thermodynamic optimization in nature has also been proposed as a fourth law, or the ecological law of thermodynamics.

## 6. Exergetics

Some good biological and ecological intuition tells us that biological systems may have many different ways of using the energy, short term or long term, of which some are more sensible to variation in external variables than others. Logically, in order to improve their chances to survive or improve life conditions in general organisms or biological systems should tend to maximize their inputs of “negentropy” and minimize their expenditures, that is their entropy production. This is the general picture but the situation may be more complicated. For instance the “successful” strategy for the bear, in the example mentioned above, was to eat as much as possible when food is available (even it may have no need at present) in order to be able to wait for an input of food next year and by minimizing the use of energy for a period.

We have also, in the previous sections, learned that energy of different forms has different quality or values, and that some of the energy forms are more valuable than others. Therefore, we should also expect life forms to keep their internal energy values on the highest possible level and for as long time as possible. Such a logic is shared with H.T. Odum’s interpretation of Lotka’s Maximum Power principle, cf. Odum and Pinkerton [22].

Within the engineering sciences a concept is found that expresses exactly the value of energy, or in other words, it accounts for the capacity of a given amount of energy to do work. This concept is called *exergy* and is defined as the maximum work capacity of energy. Implicitly, a certain part of the energy cannot be converted into work. This part is sometimes referred to as *anergy,* leading to the following expression:(16)energy=exergy+anergy
where one part—the exergy—is able to work, the other—the anergy—is not, such as the term tells.

Tracking these two parts of energy as they move through or around in a system tells how well the energy is used. A translation of the Prigoginean world-view where flows are formulated in terms of exergy may be seen in Figure 15. 

In spite of the fact that exergy is not a totally new concept, it has entered thermodynamic textbooks and papers relatively recently [159,161]. As the concept also tells us how available the energy is, another term often used is *availability* seemingly dating back to Gibbs. Therefore, the equations for exergy are often found under “availability functions” [160,205].

Exergy analysis, or second law analysis [206,207,208,209,210,211,212] is often used in attempts to optimize various engineering installations and production plants and have for some years been practiced worldwide as a common engineering practice [213,214,215]. Exergy analysis has even been carried out on societal level [216,217,218] and is proposed as a base for environmental taxing [219,220].

### 6.1. Thermodynamic Information

The exergy of a system in its simplest definition is defined as the maximum work one can get out of a system when in contact with and brought to equilibrium with a particular environment. In classical thermodynamics this will often be interpreted as what we will call a “true” thermodynamic equilibrium, but systems under such conditions or even close to such a situation are hardly relevant to biological or ecological systems. A more convenient measure, making exergy a more operative tool was presented by Evans [171,221] stating that:(17)Ex=T·I
where *T* is the absolute temperature and *I*, the thermodynamic information of the system, and:(18)$$I=\left(S_{eq}-S_{state}\right)$$
where *S_eq_* is the *entropy at thermodynamic equilibrium* (maximal entropy) and *S_state_* is the *actual entropy state* of the system. It is seen that exergy must be a positive term, that it has the units of energy, and that it is a measure of the system’s distance from some thermodynamic equilibrium. As mentioned, this is not necessarily equilibrium in the classical sense. For biological systems it is probably more relevant to consider a reference level as the surrounding environment. This may be illustrated by Figure 16. The system reaches a different distribution of microstates, *p_i_*, differing from the one it would have at thermodynamic equilibrium or the environment, *p_i_*_,0_, and therefore also has another entropy state. The entropy state reached will differ depending on the path taken. 

The above equation brings to us an understanding of system, where the more organised the system is, the more it deviates from thermodynamic equilibrium, the higher the exergy content it has. Some examples of organized system increasing in deviation from thermodynamic equilibrium are shown in Table 3.

This view can be applied to various levels hierarchical system as seen in Table 3. This is the foundation of the basic equation leading to the derivation of exergy for ecosystems carried out by Mejer and Jørgensen [37,38] (see later).

From the classical potentials above it is possible to calculate the exergy of a closed system [161], which comes to:(19)$$Ex=U-U_{0}+T_{0}\;\left(S_{0}-S\right)-p_{0}\;\left(V_{0}-V\right)$$
also called the maximum shaft/network potential of the system [159].

Again, for the open system we need to add a contribution from the chemical processes in the system
(20)$$Ex=U-U_{0}+T_{0}\;\left(S_{0}-S\right)-p_{0}\;\left(V_{0}-V\right)-\underset{i}{\sum }\mu_{i,0}\;\left(n_{i\;}-n_{i,0}\right)$$

For intensive treatments and derivation of the formulations of exergy refer to Kay [173], Wall [240] and Eriksson et al. [241].

### 6.2. Exergy Optimization

As presented earlier, the hypothesized rule or conjecture, now formulated in its widest sense, states that biological systems should tend to optimize their exergy (storage). This has earlier been proposed as an *ecological law* or 4th law of thermodynamics [7]. We will not touch further upon the relevance of the proposal here as this shall be discussed later, see Section 7.2. 

To end this presentation of the assumed basic connections between life and thermodynamics we shall try to summarize here our viewpoints on how is it possible to view life as thermodynamic structures: (1)Living systems—all biological systems as well as ecosystems are open systems in the sense that they import and exchange both energy and matter with the environment in which they are embedded(2)As the imported energy through metabolism is used for driving irreversible processes it is at the same time
(a)converted into lower quality/value energy forms exporting dissipated energy to the surrounding environment, and/or(b)built into intermediate, chemical energy compounds, thereby(3)Building up structures through processes such as auto-poiesis, autocatalysis, and self-organization driven by the energy and material gradient of the system.

Living systems become localized states which are, if not exactly at minimum, then low entropy/high exergy regions as compared to inanimate matter or composition of space in the universe, i.e., their thermodynamic state is optimized in accordance with prevailing conditions as well as both internal and external constrains

## 7. Application of Thermodynamics to Ecology

The thermodynamic approach has found a widespread application in ecology over the last decades, although the scientific importance was recognized for somewhat earlier in parts of the world where literature has not been accessible to western researchers, either due to political or language problems. Some frequently proposed understandings of hierarchies are presented in Table 4. 

It may be difficult to determine exactly where ecology in this sense begins, does it start with the organisms (at autecological level), populations or only at the level of ecosystems as a whole. Thermodynamic approaches have been applied to analyze biological systems and phenomena over a wide range of areas and at all hierarchy levels [242]. Especially, the use of entropy and information, previously mentioned, have contributed a lot by confusing and intermingled use of the concepts [16,138].

Thus, one or more of these approaches have entered for instance in physiology, especially through *network thermodynamics* [243] and Mikulecky [244,245] and analysis of aging processes, e.g., in cells [246,247,248,249]. Psychologist have also taken up the concepts in the analysis of speech [250] or other communication [251]. Meanwhile, the studies on organisms, through embryology and physiology, —on one hand serve to illustrate the principles on relatively simple systems, but—on the other hand also tell us how problematic a thermodynamic interpretation of the results may be. Hence, we will not integrate these approaches in the following treatment.

### 7.1. Entropy of Biological Systems

Measures and calculations of thermodynamic balances of biological and ecological systems have been carried out by some authors but only in a relatively few cases. A vast amount of work seems to be dominated by embryological and morphogenetic studies, as organism grow, mainly coming from Eastern Europe and Russia, e.g., Zotin and Zotina, [252,253,254,255], Gladyshev, [256,257] and Ebeling and Volkenstein, [175,176,258]. The significance to ecosystems was argued a little later through the works of Straskraba, Mauersberger [48,259,260,261,262,263], Svirezhev [264,265] and Aoki, e.g., [266].

#### 7.1.1. Entropy and Developmental Biology

Since this area is situated just on the borders of ecology it will not receive much attention here. Meanwhile, a relatively large amount of work has been carried out in the field [252] and some findings demonstrate interesting analogies to ecosystem evolution and development. What might be more important to us is that much of the theoretical background above can also be found in some voluminous treatments of the area [253,254,255]. For readers interested in a more intensive, theoretical treatment of the ideas than the version given above, much inspiration may be found here. 

In these works, entropy production has been calculated throughout the development of organisms, mainly concentrated on embryos and early epigenetic development. The observed results are found to be consistent with the Prigogine-Wiame theory for living systems [29], arguing that they through their development will move towards a state of minimum dissipation density. Meanwhile, when moving up in the biological hierarchy results for instance from population level are not always consistent with this principle as indicated by respiration data from an ant colony [267,268].

Other physiological studies establishing entropy balances based on physiological studies may be found (see above mentioned references), but will not be introduced here, except from the findings of Aoki (see the following). 

#### 7.1.2. Entropy and Organisms

Accepting the view that living systems can be treated as dissipative structures, according to the Prigogine-Wiame hypothesis [29], makes it of course interesting to establish, if possible, entropy balances for whole organisms or parts hereof.

Studies on plants, or rather parts of plants, that is leaves of soybean and bur oak [269], (unspecified) deciduous plant leaves [270], and conifer branches [271], in general show that the entropy balances in these parts of the plants are negative, although night time activity differs from that of the day [34,269]. This is maybe not so surprising when photosynthesis and thus capture of low entropy, high exergy forms of energy such as radiation is involved. Similar studies have been made where the maximum entropy principle has been used for analysis of plants and vegetation [272,273,274,275]. The quoted studies together with the technique and observations by Luvall and Holbo [276] makes up and important platform to future thermodynamic vegetation studies.

Some studies have been carried out on animals, like lizards [277], white tailed deer, [266], and even humans, e.g., [278]. The animal studies show that the net entropy flow to the animal are negative. Size may also play an important role as specific entropy production should be expected to be smaller for large animals. Calculating the specific entropy production for deer and comparing it with a lizard, shows a value for the deer of only 1/21 of the value of the lizard [266,277]. 

Studies on humans do not show such clear results cf. [278,279,280,281]. Entropy production of humans is showing a rapid increase during the first years of human lifespan, although results vary [278,281]. After the initial increase, entropy production tends to decrease with various rates and tends not to find a constant level. Results in accordance with these observations were likewise found to be valid for swine [282] and is argued to be a new “universal law for development, growth and aging of many species of biological organisms” [92,282]. 

#### 7.1.3. Entropy of Ecosystems

Studies of entropy balances have also been carried out on whole ecosystems, mainly lakes, and in particular lake Biwa, Japan’s largest and most studied lake [34,35] as well as the American Lake Mendota, one of the most studied lakes of the world. Lake Biwa is considered to be meso-oligotrophic, and Lake Mendota to be eutrophic.

Investigations of the monthly entropy balances carried out on Lake Mendota and Lake Biwa [34,35,227] showed a performance of negative net entropy flows to the lakes throughout the year. The monthly entropy production was shown to be a linear function of solar radiation. Comparing this with a more eutrophic lake, Lake Mendota, showed that eutrophication is accompanied by an increase in entropy production.

#### 7.1.4. Other Studies

Other studies show that also the net entropy flux to the atmosphere and to the surface of the earth becomes negative [31]. This potentially really allows us to view our biosphere as a local entropy minimum in the universe.

Attempts to combine the entropy view with other ecological theories, analysing the entropy in network context [283], or comparisons with the exergy approach are found in Aoki [284]. Likewise, an extension of the exergy approach combining it with Kullback’s index of information has been found [285]. Such a view makes a connection to the early derivation of the exergy concept to ecosystems as established by Jørgensen and Mejer [37,38].

Beyond no doubt this presentation is not exhaustive and many more examples are likely to be found in the future, e.g., [94]. This is valid, especially in the case of Russian and former Eastern European literature that recently have become more accessible, judging for references in the works of, e.g., Gladyshev, Svirezhev, and Vernadsky, quoted in the above.

### 7.2. Exergy and Ecosystems

The concept of exergy has been introduced for ecosystems since the late 1970′ies. It was already argued, from the beginning, that his approach is holistic, correlating to the stability of the system, in the sense of buffer capacity, and that it could possibly be a goal function to ecosystem. A continuous refinement and development of the theoretical background has been going on ever since the first formulation of such a metaphysic was stated at the end of the 70′ies and beginning of 80′ies and up to the early 1990′ies [37,38]. Somewhat later, a slightly different interpretation of exergy for ecosystems was presented by the American and Canadian researchers Schneider and Kay, e.g., [39,222,224,225]. They argued that the breakdown of the gradient of exergy imposed on the system was the major factor in determining the development of biological systems. Whereas, the distinction between the two directions for the time being may seem subtle to the reader we will attempt to come up with more explicit and precise interpretations in the following. In any case, the introduction of a distinction between the two attitudes became necessary as it correlates strongly with the previously presented discussions of differences between maximization and minimization of entropy formation and whither this should be based on entropy as an extensive or intensive variable. We shall therefore refer to the first concept as the *exergy storage* approach, and to the latter as the *exergy degradation* approach.

### 7.3. Exergy Storage

Through the early works of Jørgensen and Mejer, the thermodynamic availability function known as exergy was introduced as an indicator of ecosystem state in the late seventies [38]. Through international cooperation and projects various types of exergy expressions have been derived and tested [231,236,286,287,288]. Lately, the classical approach has been even more refined resulting in the concept of eco-exergy as a consequence to criticism received [234,235,289,290,291,292].

#### 7.3.1. The Classical Approach

The first attempts to expand the thermodynamic function exergy to ecosystems were, as mentioned, found in the late seventies and early eighties. The earliest descriptions and examinations are found in Mejer and Jørgensen [37]. The full derivation of the concept going from the classical potentials to the ecosystem level can be found in [37,38].

The derivation is carried out by putting the *thermodynamic information*, *I*, as defined above in Equation (18), equal to:(21)$$I=\frac{U+PV-TS-\sum_{i}\mu_{i}\;n_{i}}{T}$$
meaning that exergy, *Ex*, equals:(22)$$Ex=U+PV-TS-\underset{i}{\sum }\mu_{i\;}n_{i}\;\;.$$
which only differs from the above classical, engineering expression for Gibb’s free energy by the extra summation part containing the contribution from the chemical compounds or elements. In fact, this may be the closest we get to formulate Gibb’s free energy of a biological system. In should be noted that in both Equations (21) and (22) carries and implicit deduction of thermodynamic information and Exergy at a reference state, *I*_0_ and *Ex*_0_, respectively. As both these reference values by definition are 0 (zero) they have been omitted from the equations). 

Exergy was in a series of papers derived for a simple lake model and generalized to be:(23)$$Ex=R\;T\;\overset{n}{\underset{c=0}{\sum }}\left[C_{i}\ln\frac{C_{i}}{C_{eq,i}}-\left(C_{i}-C_{eq,i}\right)\right]\;$$
where *C_i_* is the concentration of a given chemical element in various compartments of the system. The equation is argued to be valid for systems with inorganic net inflow and passive organic outflow [38]. 

#### 7.3.2. Internal Exergy

The launching of another expression, the *internal exergy* must be seen as an attempt to formulated the part of the exergy caused by the ecosystem structure alone. In this examination this internal exergy fraction was assumed to be separate from an externalized part related to the external exergy functions imposed on the system, hence denoted as *external exergy*. The expression for internal exergy was proposed by Herendeen [293] as:(24)$$Ex_{intern}=R\;T\;\overset{n}{\underset{c=0}{\sum }}\left[x_{i}\ln\frac{x_{i}}{x_{eq,i}}\right]\;\;$$
where *x_i_* are the fractions of chemical elements in the compartments of the system. The expression can in fact be shown to be equal the above expression without the relations to the external. The difference between the above presented approaches was analyzed through the thesis work of Nielsen [190,230,294,295,296,297] and the variation was found to differ only slightly and in particular when structural shifts occurred in the ecosystems under consideration.

#### 7.3.3. Exergy Indices

No new concepts are introduced without difficulties and criticism, and the two previous formulations had already some, at least two, built-in problems. Both problems were related to the formulation of *C_eq,i_* (or *x_eq,i_* in the latter). In order to explain this, let us try to translate this term into ordinary chemical language. What we seek to express through this equation is: the probability of finding organic compounds, eventually put together as an organism (or several types of organisms of the ecosystem) but at a state of thermodynamic equilibrium or in the primordial soup. In the equations this in turn would need to be expressed as its “hypothetical” concentration under the same conditions. A conflict arises since no life is assumed to exist under the conditions of thermodynamic equilibrium. This creates the first problem. The probabilities are extremely low and the first half of the equation becomes dominant in the calculation. Furthermore, they were estimated to be so low, (around 10^−50^) that they were hard to accept since such values could neither be proved nor measured. 

Second, ecologists tend to evaluate the hierarchic organization of organisms in the ecosystems against each other on the basis of some intuitive understanding of complexity. We, for instance, intuitively rank the upper part of the food chain, like vertebrates, such as fish much higher than the primitive organism at its basis, which might not even be eukaryotes, e.g., phytoplankton. Indeed, this intuition corresponds very nicely with the impressions of their thermodynamic importance we would implicitly derive from the above. The more advanced an organism, the more structure and exergy it has, the higher the thermodynamic costs it has taken to build up that very structure. Meanwhile, the only way we could distinguish between the different levels of organisms in the ecological hierarchy would in this case of classical exergy concept—be to make a distinction based on the estimated values of *C_eq,i_* (or *x_eq,i_*). This was not considered to be quite satisfactory for the reasons just described, but mainly due to the problems related to the unrealistic possibility of life to exist at the reference level of thermodynamic equilibrium.

In order to meet the criticism two measures were taken. The first part of problem described is basically a problem of the reference level chosen for the ecosystem. One obvious solution was to choose the reference level of the ecosystem equal to detritus or inorganic nutrients, that is, - to view detritus or inorganic nutrients as the ultimate end and starting point of all organic matter. Most (dead) organic matter has an energy content of approx. 18.4 Joule gram^−1^ [234,235] thus giving a basic value to be used in the formulation of another possible reference level.

The idea then came by to evaluate the exergy state of the various biological compartments or organisms in the ecosystem (as compared to the previous value) by the amount of information embedded in their genetic material. This means that this new eco-exergy is believed to express the complexity level of the system [52,298]. Meanwhile, a major problem arise from this idea as total amount of DNA is not well correlated to the complexity level of the organisms. This means, that the total amount of genetic material is not directly useful for this calculation. Rather we should include only the part of the genome which is believed to be expressed during the life history of the organism. Hence, we must find ways to identify this part, which is a search process that still goes on. 

Based on these values of information found in the genome of various organisms [289,290] it was possible to establish a weighting factor, *β_i_*, to be multiplied with the biomass of each of the compartments, thus this new exergy index, often referred to as is defined as:(25)$$\frac{Ex}{R\;T}=\underset{i}{\sum }\beta_{i}\;C_{i}\;$$
where *C_i_* is the biomass, expressed as concentration, of compartment *i*. 

Surprisingly enough, in spite of the efforts and money laid out for biotechnological research, only little relevant knowledge of this type exists in current literature and textbooks. A first “rough” table summarizing the values used up to now for aquatic systems is shown in Table 5. There is hope that this table will be more complete as values are added in the future through for instance all the research connected to studies of biodiversity. New techniques are at present under development making it easier to establish this kind of estimates [236,289].

The above index also leads to another measure that recently has been implemented in order to indicate the quality status of ecosystems. The new index has often been referred to as *structural exergy,* or *normalized exergy.* More recently, the term *specific exergy* seems to have been preferred. The specific exergy is calculated as:(26)$$Ex_{spec}=\underset{i}{\sum }\frac{\beta_{i}\;C_{i}}{C_{tot}}\;\;$$

As seen the expression is divided by the total biomass of the system and therefore expresses how the exergy is distributed among the compartments of the system. Intuitively, when this index is low, high biomass combined with low exergy, this will indicate a malfunctioning or at least sub-optimal system. When the index is high, at the extreme, we have low biomass of high quality, we have a system where resources, even if scarce, are well converted into quality. For more accurate descriptions, derivations and considerations, please refer to Jørgensen et al. [234] and Bendoricchio and Jørgensen [235].

### 7.4. Application of Exergy Storage

Demonstrations of the implementation of various exergy expressions so far can in principle be divided into three categories:

(a) Observation and evaluation of the different forms

(b) Implementation as goal functions

(c) Comparisons to other ecosystem theories

Examples of applications were reviewed in Nielsen et al. [288], Marques et al., [287] and Nielsen and Jørgensen [204] and we will summarize the results here with some recent results added.

#### 7.4.1. Observation and Evaluation

The different forms of exergy, four in all, have been tested several times in connection with modelling studies of mainly Danish shallow lakes and also some European lakes and estuaries. The results showed that structural changes in ecosystems during the evolution and development of phytoplankton societies in particular usually are accompanied by an increase in exergy, although decreasing trends in the transition period are likely to occur. Furthermore, the winning species, or constellation hereof, are the ones able to demonstrate the highest exergy calculated using Equations (23) and (24) above [228,230]. Furthermore, a difference in exergy values when evaluating the behavior of two of the proposed expressions, classical and internal exergy, was only conspicuous during abrupt changes of the ecosystem.

In studies on three European estuaries, Lagoon of Venice, Italy, Figueira da Foz, Portugal, and Roskilde Fjord, Denmark, similar results were found [236,299]. The new exergy forms (Equations (25) and (26))—now often referred to as eco-exergy—were added to the investigations. Only the normalized exergy seems to deviate in its behavior from the others when compared. In a Portuguese estuary exergy clearly differs between the areas with various degrees of pollution [236,287] but reaching a maximum in an eutrophic area dominated by macro-algal blooms, which probably relates to the *intermediate disturbance hypothesis* (IDH) suggested by Connell [300] described later.

On the Italian lake, Lake Annone, it was found that the parameters found by calibration would give, not alone the highest exergy, but also result in simulations in better accordance with observation of the state variables. The case study had the aim to analyze the transformations of the ecosystem that had occurred as result of a massive fish death caused by gill infection in 1995 [286]. 

All exergy forms, including specific exergy, were analyzed on a Chinese lake, Chao Hu, and the performance of each indicator following the application of ecological engineering measure like biomanipulation experiments were examined. The specific exergy was found to be promising as indicator of the distribution of biomass among the compartments, implicitly telling how well the ecosystem is structured [238,239,301].

In an elegant study, calculations of growth of macrophytes based on allometric relations derived from literature were coupled to a calculation of which of a library species would be predicted to perform demonstrating a maximum exergy under prevailing conditions in the lagoons of Venice. The calculations were used to predict the distribution of various macrophyte species in the lagoon which in turn could be compared with the observed distribution. Results showed a high accuracy, a more than 85% hit, in the predictions [302].

#### 7.4.2. Goal Functions

Exergy has also been implemented as a mathematical goal functions in simulations with the purpose of improving the predictive value of existing models. This type of application has only been carried out in a small number of cases [190,228]. This is due to the fact that present modelling tools for computers are not well fitted to this type of modelling which involves continuous optimization and thus manipulation of several parameters in the models. 

Meanwhile, results show that it is possible to manipulate parameters in accordance with the maximum exergy principle, thereby hypothetically imitating the adaptational processes taking place under natural ecosystem development. In brief, results indicate that an allowed adjustment rate of the parameters of 10% and an adjustment interval of approx. 10 days seems to be a good starting point for future attempts in the area. The results of this type of optimization, take a good deal of both mathematical and programming skills and requires a deep knowledge of the algorithms, i.e., how they actually work, before they can be used properly. 

#### 7.4.3. Comparisons to Other Ecosystem Theories

One of the first optimization results [228], also indicated a possible connection to the stability of the system and maybe even to chaos theory. By implementing optimization of exergy to the parameters of the system, leading it through a transition induced by changes in fish populations corresponding to empirical observations, the fluctuations were damped considerably. Meanwhile, carrying out simulations maintaining the initial set of parameters the transition led to a highly fluctuating and unstable system. To a certain extent the system was demonstrating *deterministic chaos* in fulfilling the simple requirement of sensitivity to initial conditions and exponentially increasing deviation (Jørgensen, personal communications). Whether this means that exergy optimization helps systems to avoid chaos, i.e., staying on the safe side of “the edge of chaos”, remains to be clarified.

Ecosystems which are from time to time exposed to disturbances moderate in size and frequency are sometimes reported to be systems exhibiting an elevated or higher diversity than expected [300], and this phenomenon is often referred to as Intermediate Disturbance Hypothesis (IDH). Studies that correlate IDH with exergy have been carried out on Lake Balaton, Hungary [231], and in connection to the project on estuarine systems as indicated above [236].

Furthermore, it has been shown that exergy correlates well with several other of the proposed indicators or orientors of ecosystem function found in the current literature. (see Table 3). Analysis performed on particular systems or system types like an aquatic network as shown in Figure 17 are found to show a good correlation with the concept of *ascendency* proposed by Ulanowicz [5,16], see studies by Jørgensen [303], Jørgensen and Ulanowicz [304], Jørgensen and Nielsen [305], Christensen [306] and Salomonsen [307]. 

Likewise, it has been shown that changes in flows predicted to improve the thermodynamic efficiency of the ecosystem network will also eventually contribute to a positive change in ascendency of the system [232]. There have been also some indications that exergy may find a correlation with the concept of *utility* and *indirect effect* as introduced by Patten (for extensive overview see Higashi and Burns [4]. Meanwhile, this issue awaits further investigations.

### 7.5. Exergy Degradation

An approach very similar to the one described above has been taken by Schneider and Kay in a series of papers [15,39,40,222,223,225]. Meanwhile, whereas the two approaches seem to be similar from a superficial point of view, some fundamental discrepancies exist. Even if Kay and Schneider in many ways in their presentation seem to be fundamentally in accordance with the conjectures presented above—in particular their interpretation of the importance of thermodynamics to life and evolutionary processes—a marked distinction becomes apparent when it comes to a discussion of how this manifestations of the second law comes by. 

The hypothesis presented in their works is proposed to resolve Schrödinger’s “dilemma” with negentropy and reconcile it with the approach to living systems as dissipative structures [74,78] while at the same time being much influenced by views of maximum entropy states or formation. The thermodynamic view of Schneider and Kay [39] builds on thermodynamic views from theoreticians throughout the century, like Carathéodory [308], Hatsopoulos and Keenan [164,309], Kestin, [182] and Jaynes [36].

First of all, the authors refer to the fact also presented earlier, that the concept of entropy is only clearly defined at conditions close to (thermodynamic) equilibrium. Thus, they are really in favor of a formulations of the second law under far from equilibrium (FFE) conditions which is freed from the concept of entropy and therefore a consequence be applicable to FFE-conditions. Such formulations may be found in the following restatements, quoted from Hatsopoulos and Keenan [164] and Kestin [175] quoted from Schneider and Kay, [14] that unifies the two thermodynamic laws:


*“When an isolated (sic!) system performs a process after the removal of a series of internal constraints, it will reach a unique state of equilibrium: this state of equilibrium is independent of the order in which the constraints are removed”.*


This statement is presented as valid to “closed isolated” (sic!) systems only [39]. 

As mentioned, another core point for the understanding of this approach is the acknowledgement of living systems as dissipative structures. The structures realized appear as a consequence of thermodynamic flows, either of energy or materials imposed on the system. It is further argued that the structure(s) which emerges will be the one(s) that facilitates the use of the gradient, i.e., dissipation (see later, e.g., the “restated second law”). As an example, and partly also proof, the authors mention the Bénard-cells. In this type of experiment a fluid is kept between two surfaces of which the lower is connected to a heat source. Exceeding a certain threshold value of the heat flux to the system, convective cells exhibit a conspicuous, often hexagonal pattern, referred to as Bénard cells, will form in the fluid. The cells help to move heat from the bottom to the upper surface in a faster manner. The patterns occurring facilitates heat transfer and entropy formation by removing the (thermal) gradient in a faster manner than normal convection. The experiments and the authors’ representations are found in details in Schneider and Kay [39].

According to Schneider and Kay [14] it was Kestin who took this approach to systems a step further by showing that, at the “equilibrium” state, systems are stable in the Lyapunov sense. This proof bears implicitly the conclusion that the system will resist to move away from this “equilibrium”. (“Equilibrium” here seems to be used in the sense of dynamic equilibrium or stationary state not necessarily close to thermodynamic equilibrium conditions).

The above leads to the following restated second law:


*“As systems are moved away from equilibrium, they will utilize all avenues available to counter/resist the applied gradients. As the applied gradients increase, so does the system’s ability to oppose further the movement from equilibrium”*
[39]

An approach like the exergy degradation clearly sets focus on the functionality of the ecosystems. Therefore exergy has been proposed to be an important factor (orientor) in the assessment of the integrity of ecosystems [200,222,236,287,310,311].

According to Kay [222] an ecosystem, if following natural evolution, “will move away from local thermodynamic equilibrium, (i.e., a steady/stable, but yet dynamic state (authors comment) along a stable *thermodynamic path* in phase space” (see Figure 18 compare Figure 12 and Figure 13). While following a given path, new possibilities of stable/steady states may be found i.e., the path may split in several branches (cf. Gould’s punctuated equilibria [196]).

As the system evolves it will tend to develop towards what is considered to be an *optimal operating point*—that is to break down, degrade the exploitable gradient(s) of exergy as much as possible—which most likely also represent some stability, e.g., [88,312].

The system may be moved away from the path and the optimum operating point as a consequence disturbance of different magnitudes, i.e., changes in external factors, like climatic changes and changes resulting from human activities. When external factors are changed one might expect one of several things to happen (see Figure 19). To simplify the views, (1) the system is able to resist the changes and stays at its optimum operating point, or it may lower its functionality only slightly, i.e., stay close to 1. (2) The system is not able to resist changes but collapses and moves away from the optimal operating point permanently. It may stay at the same branch and reach another optimum operating point (3a) through a bifurcation point (2), or it may switch to another branch with a totally different operating point (4, through the states 2 and 3b). For a more detailed description and examples, please refer to Kay [15]. 

### 7.6. Results of Exergy Degradation

When it comes to actual results, the exergy degradation approach inherits exactly the same problem of measuring and the existence of adequate data for the calculation of this property. Meanwhile, some attempts to estimate the exergy degradation from various types of ecosystems have been made and the results have been presented in several papers. The results are quite promising in particular to the evaluation of ecosystems at macroscale. 

The techniques used in estimating exergy degradation in ecosystems (vegetation) have been developed by Luvall and Holbo [276,313,314] in the late 80s. Explained in a simple manner the exergy gradient imposed on the ecosystems in form of solar radiation is used by plants for photosynthesis, which is a dominant or primary activity of any ecosystem. The process of photosynthesis is, in turn, followed by evapotranspiration, an activity generally recognized to result in a cooling of the system. Evapotranspiration and cooling thus becomes a measure of the activity and maturity of the ecosystem. The more advanced or mature the ecosystem is the more evapotranspiration takes place. The more evapotranspiration, the relatively greater the cooling of the system will be. So eventually, the cooling observed becomes a proxy to activity and exergy degradation. 

It is theoretically possible to estimate this cooling of the ecosystem by remote sensing measurement and this basically is what has been done with the application of the technique. The detailed descriptions of equations used may be found in Schneider and Kay [39].

#### 7.6.1. Remote Sensing, Global

In one case it has been possible to estimate the exergy degraded by some of the larger ecosystems of the world, the Amazon, central and eastern United States, Asia and Sahara, by data from satellites, i.e., remote sensing data.

The observed activities behave as expected and show that the more advanced systems the higher the evapotranspiration is. While the Amazonian rainforests are able to absorb the equivalent or slightly less than Sahara, much of the incoming solar radiation is used for production as indicated by the evapotranspiration data (70%) compared to 2% of Sahara. As a consequence, of the evapotranspiration the more mature ecosystems, e.g., the rainforests, may lower the temperature with as much as 25 °C compared to surroundings. For more details readers are kindly referred to Schneider and Kay [41]. 

#### 7.6.2. Landscapes and Regional Scale

Similar results are found at a comparatively smaller scale, the landscape level. These measurements are here carried out by the use of a device called a Thermal Infrared Multispectral Scanner (TIMS) which can be mounted on a smaller aircraft. 

Data were collected by flying over a landscape in Western Oregon, USA, thus covering different types of the vegetation in the area, spanning from a quarry to a 400 years old Douglas fir forest. The data results are consistent with the above observations and describes the highest exergy degradation and the highest cooling to take place in the most mature of the ecosystems, i.e., the forests. At the lower end of the spectrum the quarry and a forest clear cut are found. 

The results clearly indicate that the measurements are able to capture some important factors in the ecosystem—like for instance primary production activities—such as providing estimate of the exergy degraded by them. To a certain extent, they still leave us with the problem of what happens to the system as this exergy is degraded. Clearly, this is illustrated by Jørgensen [7] see following section. In short, the landscape data obtained in this manner show that an increasing amount of structure expressed as biodiversity can be maintained at the same level of exergy efficiency that is around 70%.

### 7.7. Storage or Degradation

Many of the arguments from Schneider and Kay above in general points back to the fact that they are derived from a maximum entropy viewpoint, an analogue of Jaynes and Swenson’s maximum entropy formalism [36,315] (see Levine and Tribus [316], Martyushev and Seleznev [42] for reviews of the area). These views seem to be applicable and valid as interpretations of closed systems, but when it comes to open systems, especially when reaching steady state, the approach becomes insufficient. Ecosystems seem to do something more than just maximize the exergy degradation as also more and more structure is emerging as a result of this exploitation of gradients.

Ecosystems in early stages perform in a manner that could indeed be interpreted as an entropy maximization. When the ecosystem is young or immature and resources, energy and mainly matter, is abundant the species or composition of species with the highest capacity for growth will win and take over. This in turn will lead to a sequence of ecosystems (along a thermodynamic path, see previous) where new species or trophic structures will replace each other when new conditions are met.

In the immature phase, growth of the ecosystem and its components is dominated by capturing resources, whoever comes first is first and takes as much as possible of the available resources, at least until Liebig’s law [317] is brought into play. We know this situation well from ecological concepts such as pioneer or opportunistic species. The system is not in a long-term stable situation and there is never time for the minimum dissipation principle to take over, say to become dominating.

Meanwhile, as resources get scarcer or even limiting due to the intensive growth, the only way of getting access to more resources is by optimization of the system. This “fine tuning” of parameters or adaptation as we usually call it in biology—tends to improve the efficiency of the processes, which in the thermodynamic sense corresponds to minimum or at least lowering dissipation, i.e., an optimization in accordance with constraints rather than a maximization or minimization. There may be other constellations of species, i.e., other societies or trophic structures, that may perform with a higher efficiency. Meanwhile, such better states may be inaccessible and the ecosystem is in a lock-in situation. That is, the only way of getting to other stability points must involve either a partial or maybe even total destruction of the ecosystem, like it is the case of catastrophes, chaotic oscillations, or Holling cycling [310,318].

This is partly confirmed when plotting the remote sensing data of Holbo and Luvall, [276,313,314] as a function of the exergy storage of the ecosystems. In this case we are interested in the exergy efficiency, expressed as percentage of incoming exergy utilised by the system (see Figure 20).

When estimating exergy efficiency of the above systems, the exergy captured and degraded is found to raise rapidly in the initial phase of ecosystem development. At a level of surprisingly low maturity the exergy captured is levelling off, i.e., the efficiency is not growing any longer. This takes place in spite of the fact that the ecosystem holds higher and higher biomass and thus has an increased exergy storage. On the plateau of the curve forested ecosystems are found, which is consistent with the *climax society* principle of E.P. Odum previously mentioned.

The systems, though, do vary in structure and complexity, the Amazonian rainforests assumed being the most complex system of them all. So, this observation indicates that while the exergy efficiency of the system is similar it is at the same time able to maintain a much higher or more complex structure. For instance, much more biomass and diversity at the same “costs” of maintenance. Thus, the climax societies are performing in a manner consistent with the minimum specific dissipation and the maximum exergy storage principle.

## 8. Discussion and Future

From our experience and discussions, we have had throughout the latest years, the obstacles to the acceptance of interpreting ecology within a thermodynamic framework are many. Meanwhile, although many in number, the problems may be grouped in three distinct types, which briefly will be laid out here:(a)Problems related to science of physics - the science of thermodynamics and particular its extension into the far-from-equilibrium domain of conglomerate systems is still a relatively new discipline and in many ways in opposition to the Newtonian and determinist worldview still held by many scientists. As a consequence, many discussions are still taking place within the area.(b)Problems of transfer—whenever a scientific theory is transferred (reduced) to another area problems are to be expected. Does the theory, or the transfer of it, hold at all, for the whole set of systems or for parts of it, i.e., is the transfer to new conditions or domains valid?(c)Problems of application—after theoretical transfer problems of practical application appear. This in brief deals with both problems of measuring as well as how to proof the validity of such theories after transfer. Insofar, we must take much of the above statements as conjectures although much evidence of at least some important thermodynamic features of ecosystems has been gathered.

It is clear that new, adequate and operational definitions are needed which may describe the state of systems and their dynamics in objective manner, as well as inclusion of description on how they deviate from traditional definitions.

All of these problems are in principle found and to be expected when reductions are taking place within or between major areas in natural sciences [319]. Some will even argue that the criticism held within the area of thermodynamic interpretation is so severe that it will eventually lead to a shift in paradigm in the sense of Kuhn [320]. It is still too early to address this point but a first attempt to examine the consequences of reductions is given in Nielsen [16]. Today’s interpretations of living systems within a thermodynamic framework in general offer solutions to essential problems raised by traditional science, i.e., explanations in terms of causality in a better way than many established theories. This does not necessarily mean that the thermodynamic view replaces the traditional *normal science* (as Kuhn would have described it) but merely that it offers something new in addition to it. In the following, we will have a closer look and summarize the fundamental problems.

Thermodynamics is a relatively “young” discipline within physics as compared to geometry and Newtonian physics and in Kuhnian sense it has still not established into a totally fixed normal paradigm, in particular not when it comes to far-from-equilibrium issues. It offers a new worldview able to help us solving problems where the traditional Newtonian worldview is not sufficient. As a consequence, far from equilibrium thermodynamics is still under development, which is really the case with the extension of its applications to biological and ecological systems. At the same time, thermodynamics is a hard discipline to get at, even for physicists, and new areas of applications are continuously occurring.

As a minor example of problems of this new science, one just has to look at the inconsistencies in the use of terminology, e.g., isolated, closed or open systems, that is found among the many textbooks on this topic today. Not to mention a concept and word like exergy which has only been accepted very recently although similar ideas have been around from the start of thermodynamics.

(a)Most important to establishing a connection between thermodynamics and biology seems to be the necessary extension of the validity of thermodynamics into far-from-equilibrium conditions. The traditional point taken, stated in a very simplified form, would argue that thermodynamics only deals with ideal gases at conditions close to thermodynamic equilibrium. Whatever variety, or nuances, of this attitude will be taken, it will bring the transfer and application of thermodynamics into deep trouble. If one stands hard on the point that thermodynamics as science is valid only to “ideal gases close to (real) thermodynamic equilibrium, not only will the situation in biology and ecology be in deep trouble, so would a large part of the physical and engineering sciences as the universality and role of the second law together with its penetration into all other physical disciplines vanishes.

We will be forced in the future to answer the question of how far we can take thermodynamics. How far away is far-from-equilibrium? Are there objects so far from equilibrium that interpretation within a thermodynamic framework does not hold any longer? Is it still thermodynamics or something else, i.e., a whole new scientific discipline? And do we agree that we will be able to transfer the discipline to those conditions, such as living systems, composites/conglomerates, complex adaptive system, etc.? Questions like these are already inherent in physics. But they will need to be answered even more explicitly or it will be necessary to reach at least a certain consensus before the area will and can be consistent enough to make a “sound” entrance point for biological interpretations.

(b)With the last points we implicitly address the problem of transfer to biological sciences and also to ecology as presented above. At this point it should be clear that not all problems come from the transfer alone, they existed already.

When it comes to the application to biology or even ecology, due to the unsolved problems referred to above, one must face problems. Clearly, many of the problems recognized today refer to many of the above issues. In transferring the concept to the biological area, the danger exists that views and problems from the physicist world are transferred too, and one might ask the question to what extent this affects the discussions today. Scientists working with physical issues, where the maximum entropy paradigm [316] is the dominant and tacitly accepted framework of interpretation (i.e., current paradigm), are likely to maintain that view during the transfer. Likely, the same will happen when scientists approach the problem from a more Prigoginean angle, i.e., from the dogma of minimum (specific) dissipation. It remains to resolve the problem exactly what interpretation fits biological systems better. It seems possible that the two views are not necessarily exclusive and that both may be important in getting the full overview of the evolution of ecosystems as indicated above, e.g., Aoki [92].

It remains to be stated that biological systems, spanning over the range from cells and organisms to the biosphere must all be classified as open systems, although in some interpretations the biosphere is classified as (quasi-) closed for convenience, thus neglecting inputs of materials from the atmosphere. The importance to exergy of the various system’s components may differ throughout the biological hierarchy, with pressure and temperature being important at lower scales, whereas the chemical, material fluxes are by far the most dominant at ecosystem level. This does not exclude the possibility that pressure and temperature are not important factors and that the portion of energy is insignificant. Only, at the ecosystem level, they belong to the abiotic regime, in modelling referred to as forcing functions. Weather conditions in general, like precipitation, movement of water on Earth, like the Gulf Stream are driven by temperature and pressure gradients and connected with big powers. They are a part of the prevailing conditions of the ecosystem. At ecosystem level, all other organisms than the homoeothermic animals (mammals) are dependent on inputs of heat to maintain activity. 

Related to the problem of transferring thermodynamics—and at its core—lies the issue on what to choose as of reference level or dynamic equilibrium state of the surroundings. According to one definition, exergy is the maximum work that can be extracted during the process of bringing the system to equilibrium with its surroundings or environment. Clearly, this does not imply that equilibrium is necessarily interpreted as being “true” thermodynamic equilibrium. But then, what is the proper equilibrium state for a far from equilibrium system? Many suggestions have been made—like concentration of nutrients of the Oparinian sea, most oxidized states of inorganic nutrients, etc. Most likely this is of minor importance, one just needs to use the same reference level if one wish to compare results.

(c)This brings us back to yet another problem in the application of the exergy principle. None of the above presented approaches is able to measure entropy, exergy or any kind of thermodynamic balance, directly. We have no entropy syringe or exergy meter to put on our system. This means that we are not able to fulfil the Cartesian demand of “making everything measurable”. In short, we will be forced to work with inductive or abductive based methods. Except, if we accept indirect measurements, calculation or modelling as valid methods for this purpose, which seems to be our only way out of this dilemma at the moment.

As part of this discussion, the issues raised around the calculation or choice of a proper reference level is in particular valid as such values are to enter the equations anyway. The values used only need to be used in a consistent manner. In the case of exergy storage, for (classical) exergy, the level was chosen to be a dead primordial soup, in principle inorganic nutrients, but still with a probability of organic lifeforms to exist. As an answer to criticism, in the latest form of exergy (eco-exergy) this level was chosen as detritus or dead organic matter. In the case of exergy degradation measurements, the black body radiation from space is used as a reference.

Thus, the choice of reference level differs between approaches but will also likely have to differ between systems. A eutrophic aquatic system could possibly have another reference level than an oligotrophic system. A terrestrial system will most likely have another reference level than an aquatic system, and concentration will be expressed in units based on not per volume water, but per volume of soil. Terrestrial systems may have the composition of earth’s crust as reference state, aquatic systems will have the average composition of the world ocean or of lakes as reference. This difference between the various reference levels may be illustrated by Figure 21.

Meanwhile, the difference in exergy values calculated—which may be found as a consequence of choosing between the various possible levels—is considered to be small as most of the exergy in the systems is made up from the fact that they are living and only a minor part of the exergy state stems from choice of reference state made. In other words, the difference in calculated exergy value of an ecosystem, which may result as choosing between Oparinian sea, most oxidized states of inorganic nutrients or detrital is only a minor. What counts is that the ecosystem is alive and far from these equilibria.

The debate about exergy storage and exergy degradation may turn out not to be an unsolvable problem. Much of the discussion for sure lies in the distinction between whether one talks of entropy (as state), entropy production or specific entropy production. During evolution, structures form and grow accompanied by an increase in entropy production as a result. Continuous optimization of the system following a minimum dissipation will occur at any time in an attempt to minimize the costs of the structure. Meanwhile, the contribution of this process may vanish in the realms of the overall entropy productions as structures during a growing, developmental phase never manage to find a balance with the environment where this process can take over. Growing is a highly unstable situation for the ecosystem and the system will have to come close to (Lagrangian) stability for the minimum specific entropy production principle to become dominating. The two views seem to have a close linkage and might very well show to be different aspects of a unified principle.

Finally, the thermodynamic framework seems to provide a very fruitful understanding of the interactions and relations between human society and the environment. It was Georgescu-Roegen [321,322] who first pointed out that the development of our societies was at the end constrained by resources and thus thermodynamics. Associated with the increasing use of resources follows losses often referred to as pollution which is a direct consequence of the second law and can be equivalenced with dissipation, e.g., [57,323]. But whereas nature seems to perform in a “sensible” way, saving resources through for instance minimum dissipation, society does not. In fact, our societies follow trends much different from the ones observed in ecosystems and nature [324,325]. Only few other attempts to understand the whole society and the environmental problems we are facing today and linking it to an interpretation from a thermodynamic view point are known, like presented in the works of Rifkin [326] or Tiezzi [94,327,328]. It may well be through this close connection that thermodynamics will find its way into ecology. 

## 9. Summary and Conclusions

Having accepted the above problems and their possible need for further clarification in the future it seems to be possible to reach some conclusions. Interpreting ecosystems and nature within a thermodynamic framework has already come out with results that make it worthwhile to continue the work.

The future will possibly bring many new tests of the possible links between thermodynamics and the development and behavior of ecosystems. Until now, the efforts have been dominated by aquatic studies due to historical and methodological reasons. But the few preliminary results in the terrestrial area will for sure have more to follow. In a distant future we might be able to use satellites for continuous and instantaneous monitoring of ecosystem states, and possibilities today exist to observe diversities, age distributions and activities of terrestrial systems. When used together, these methods will for sure put some light on many of the issues, uncertainties and questions raised in the above. Today, the investment costs are argued to be high, but the value of this method may eventually turn out to be of greater value to us in terms of improvements in management, thereby increasing ecosystem and human health.

It has to be stressed, though, that interpretation within a thermodynamic framework, like exergy, is but one out of many approaches that may help us to reveal more about the secrets of nature’s growth and development. Other approaches, like network derived techniques from Patten and Ulanowicz are equally valid, complementary techniques sometimes even consistent with the above need for assessment and interpretation in the future. Therefore, monitoring frameworks building on several of these parameters are likely to occur, e.g., the comparisons between eMergy and exergy started recently.

To strengthen the values of these techniques more cooperation within the established ecological society is needed. This cooperation is mandatory if an adequate ecological answer is to be given by our scientific society, fulfilling the wishes of politicians and managers, to settle scientific definitions of sustainability, integrity and ecosystems health given as examples above. Only, recognizing the responsibility to give precautious answers to the many challenges and problems we face today becomes of still increasing importance to achieve sustainable existence of both nature and society. 

So, “Quo vadis” ecosystem thermodynamics or thermodynamic ecology? Over the years where the authors have been concerned with thermodynamic studies, mostly from the ecosystem side, some issues have appeared which need further attention. These issues mainly deals with bridging the apparent gap between thermodynamic approaches in physics and ecology. The introduction of thermodynamics into ecology has mainly had two types of reactions. In general, on one side a strong opposition from the ecological society often connected to accusations of teleological traditionally reputed by the biological societies, probably because of the use of words like goal functions and orientors which has often been used in this context. On the other side, a more positive reaction has been experienced from engineers and physicist who seem to welcome such attempt as it places biological systems under their normal paradigm. In the first case, as a result it has often been difficult to convince ecologists to plan empirical activities in a manner that allows to confirm or falsify forecasts based on a thermodynamic understanding about ecosystem behavior. In the second, the positive reaction has led to an almost too positive acceptance where falsification has been seen as unnecessary. Hence, some areas have received too little attention with respect to real scientific verification. As a result, there is still much work to be done in the areas of clarification of definition of thermodynamic variables and their behavior, i.e., ontological and phenomenological issues, respectively—all under the very far from equilibrium states represented by ecosystems. 

In addition, one overall observation can be made which may turn out to be important. Namely that ecologists in general seem to have taken an approach of optimization of (Gibbs) free energy or exergy states following Lotka, Odum and Pinkerton, and Jørgensen, while engineers and physicists when working with ecosystems tend to work from the angle of maximizing entropy or entropy production [42,43,329,330].

Meanwhile, from the experiences and knowledge acquired with empirical data gathered hitherto, it is possible to point out some essential fields which should be subjected to further examination and clarification. It is even possible to establish some conundrums that research initiatives should strive to verify or falsify. To facilitate the points made in the following readers must implicitly accept that thermodynamic laws are valid to and can be applied at any level in the biological hierarchy.
(a)Thermodynamics in organizational levels and hierarchy perspectives

First of all, it must be clarified what are the hierarchical relations—in time and space—and what are the exact meanings of a concept of entropy at any of these levels. At the lowest end of the hierarchy the resulting conceptualization is likely to come close to that of thermochemistry and thermodynamic evaluation may reduce to questions of either Gibbs or Helmholtz free energy of the systems. As soon as we get to higher levels and work with highly conglomerate and hierarchically embedded systems we need to determine what precisely we mean when we describe them in terms of entropy [16,331]. What is the actual meaning of life in these terms, the difference between live and dead organic material (the case use by Tiezzi [94]), relaxation times, time-space relations at various levels, etc. (b)Thermodynamics of Earth and the biosphere

New research should be open to a situation where it might be the case that relations and definitions will change when moving around between hierarchical levels. In particular when it comes to upper levels of the hierarchy and reaching large climatic regions or the planet as a whole. The planet is driven from solar radiation and is able to establish a huge system with a relatively stable thermodynamic disequilibrium [329,332]. As such, all processes in ecosystems are in a thermodynamic sense bottom-up driven by photosynthesis. The rest is about constraints [138].

Physical powers and thermodynamics together with geological relations [333] are responsible to shape a planet with highly variable life conditions, different levels of solar radiation and temperature, precipitation, etc. [17,18,334,335,336,337,338,339]. This makes it possible for many subsystems in time and space to exist that exhibit large differences with geographical position, yearly and diurnal variations. Each of them has its own specific equilibrium adapted to general conditions. (c)Entropy production vs. entropy state, exergy storage

In order to do this, and to open up discussions which aims at creating a more precise layout of a proper research agenda, or even an evaluation scheme applicable to such types of systems—with complex, hierarchical and adaptive properties—other things need to be clarified. For instance, we need to state whither we are working with entropy as indicators of states or if we are dealing with the entropy produced during processes. In addition, we need to state if we work with the concepts as extensive or intensive variables, as well as exact time and scale dependencies, for instance derivatives. 

The discussions around “maximum” entropy production vs. exergy “maximization” may well be resolved by specification of these points. As a conundrum it is possible that entropy production of ongoing processes increases while structure size increases, entropy state deviates increasingly from equilibrium through a continuous series of states which may all be considered to represent non-equilibrium steady states (NESS) [340]. The word “maximum” is used here in quotation marks as it is possible, that this type of systems are so dependent on outside constraints as co-determinants of evolution and development, that their final state may not be that of a true maximum but rather a state which demonstrates an increased entropy in compliance with the interactions between internal and external constraints, i.e., an optimum under the given conditions.(d)Thermodynamic synergism

The interactions—and as another conundrum—which most likely increase synergistic behavior between abiotic and biotic parts—should receive increased intention. At the macroscopic level an interaction exists between the climate shaped by physico-geographical factors (geographical positions such as latitude, soil texture, color and shape) and the prevailing ecosystems at each climatic belt. A dialectic interaction should be recognized where the physical and biological forces interact and together make up what is sometimes recognized Earths critical zone. On one hand we have thermodynamic forces determining temperature and rain and at the same time the activities of biotic components, vegetation in particular, which allows for the existence of a local climate of the systems which deviates strongly from the conditions which would have been observed had it been for the physical powers alone (e.g., [313,314]). 

Thus, thermodynamic relationships are believed to determine biogeochemical cycles and activities in general [341]. The same is valid to other sub-systems such as the seafloor, where systems may be viewed as one dimensional, and where arrangement of bio-geochemical balances and biotic communities of microbes are arranged as results of gradients imposed on the system. Such an organization has been proposed as related to a state of maximum entropy production [342,343,344] and to predict biogeochemical pathways [345]. In short life interacts with geology and vice versa [87,329,337].

Meanwhile, whereas this is the case for the deep floor or eutrophic areas things looks differently when organisms are inhabiting the sediments and they are exposed to the process of bioturbation. The one dimensional arrangement may according to Aller [346] be seen as a thermodynamic organization in accordance to the energetic outcome of processes in terms of ΔG. When higher level organisms like invertebrates interfere with the system, processes speed up and the organization of the whole gets similar to a “model of spaghetti” (Aller, pers.comm). Again biotic processes are interacting with physics and chemistry to determine conditions of the environment.

Most examples in this area come from aquatic systems, also some indications of similar mechanisms in terrestrial systems exist, for instance from studies on rainforests and other tropical systems as seen in the following.(e)Ecological time-space thermodynamics

As mentioned in the text we have excluded a relatively large set of papers dealing with functions -isomorphic to entropy and ecological diversity measures—used to describe distributional patterns in particular vegetation analysis. This area deserves further attention. Again, as a conundrum, the distributional patterns, in both 2- and 3-dimensional space may emerge as a special type of organization that leads to higher levels of exergy imports from solar radiation, higher levels of exergy storage, increased deviation from a dynamic thermodynamic equilibrium state, as well as increase in entropy production. This process should evolve consistently over evolution and development of the ecosystem in time and space. Several papers exist to support this theory for instance through the works of Dewar and colleagues [272,347,348]. Such analyses might well benefit from several new remote sensing techniques (e.g., Lidar and laser) for observation and measuring of biodiversity, primary production activity, evapotranspiration [276] and coupled to works on physico-geographical relations like [273,274,275,276,277,278,279,280,281,282,283,284,285].

Having said this, we believe that the above issues need to be resolved in a close interdisciplinary setting between physicists, engineer, technicians and general ecologists and ecosystem theorists. Establishing such a project will face challenges not only in making the ends of various involved disciplines meet, but also in terms of organizational and financial issues. It might be a tedious task, but is necessary if we are to keep our planet in a safe and sustainable condition for the future.

## Figures and Tables

**Figure 1 entropy-22-00820-f001:**
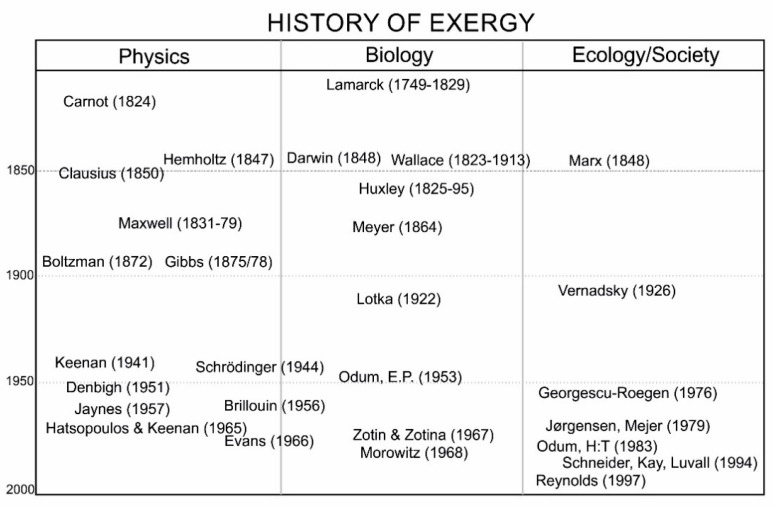
Overview over some historical events leading to the application of thermodynamics and exergy into ecology. The scheme has been divided into three areas, (1) one for the development of thermodynamics within physics, (2) a second line linking thermodynamics to biology, and (3) a third line showing important events in the development of the relation between ecology and society as we see it today. An attempt has been made to place authors in accordance with their respective areas of research efforts with indication of approximate time of major contributions.

**Figure 2 entropy-22-00820-f002:**
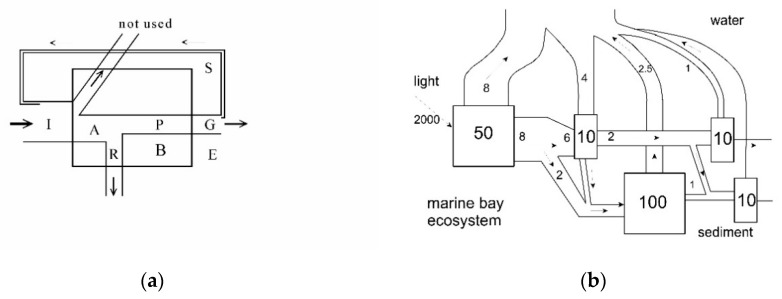
(**a**) A universal energy diagram according to E.P. Odum [2,168]. The components are: ingested energy, I, energy not used, NU, energy assimilated, A, production, P, respiration, R, growth, G, energy stored, S, and energy excreted, E. All functions carried out by biomass, B. (**b**) An energy flow diagram for a marine bay ecosystem showing the energies flowing through the grazing chain in the water column and entering the sediments, respectively. Both diagrams have been redrawn and modified after Odum [2].

**Figure 3 entropy-22-00820-f003:**
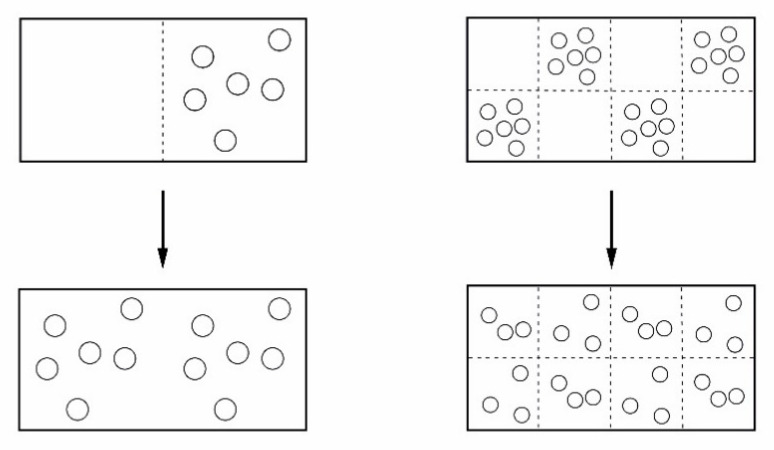
Two systems, the one more complicated than the other, both moving towards thermodynamic equilibrium, i.e., a state of more equal and more probable distribution, ending in a state of *maximum randomness.* Thus, entropy goes to maximum as elements are reaching the distribution of *highest probability* as dictated by the second law of thermodynamics, as is the situation for an isolated system (figure oversimplified).

**Figure 4 entropy-22-00820-f004:**
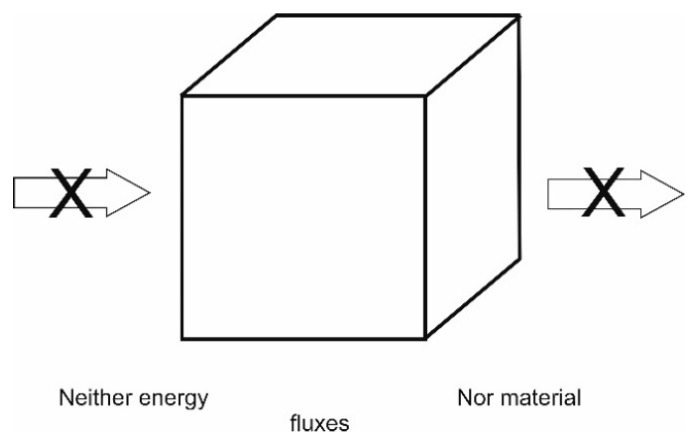
An isolated system has a boundary towards its surrounding environment, which is totally closed to exchanges of both energy and materials between the two compartments.

**Figure 5 entropy-22-00820-f005:**
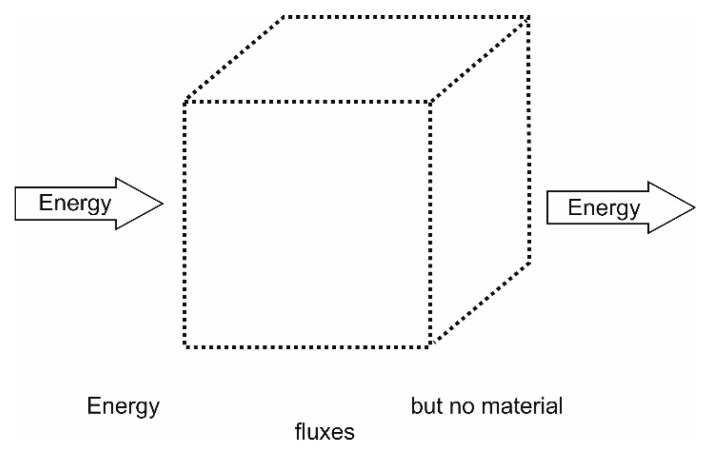
Closed systems have boundaries which are open to and may receive or exchange energy fluxes. At the same time, the materials potentially enclosed in the system at initial conditions must remain constant. Meanwhile, this still leaves the possibility of the elements to be structured or (self-)organized in various, more or less “sensible” ways according to other physicals laws, like in it is the case with some physical systems like the advective Bénard-cells mentioned in the text.

**Figure 6 entropy-22-00820-f006:**
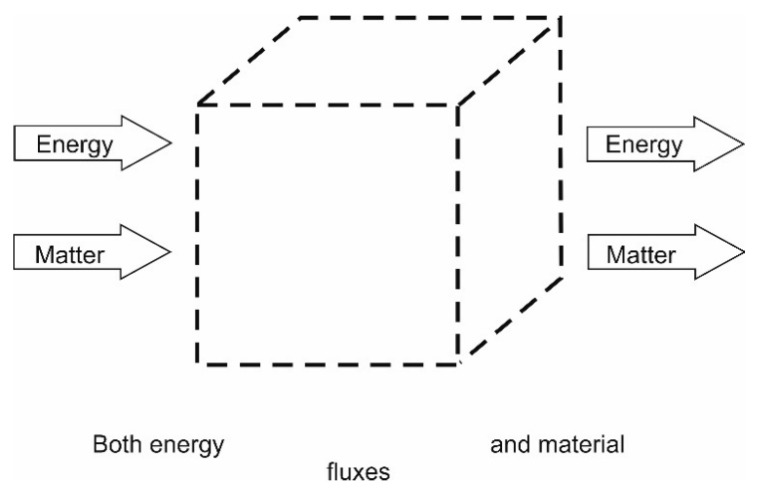
Open systems are open to both energy and material fluxes. They may use the energy and material fluxes received to build-up and organize matter or compositional elements, distributing them in still more advanced patterns and even have the capability to grow in size. Energy in general needs to leave the system as dictated by the second law always, e.g., the heat formed by dissipative processes, for instance through metabolism. The dissipated energy must disperse to the environment and this surrounding reservoir in turn must be able to tolerate this [138]. That matter leaves is not a necessity unless we for instance think of degraded compounds which otherwise might be harmful to organisms, c.f. the role of kidneys. Meanwhile, for many biological systems exchange pattern will be determined by ecological roles or by forcing functions.

**Figure 7 entropy-22-00820-f007:**
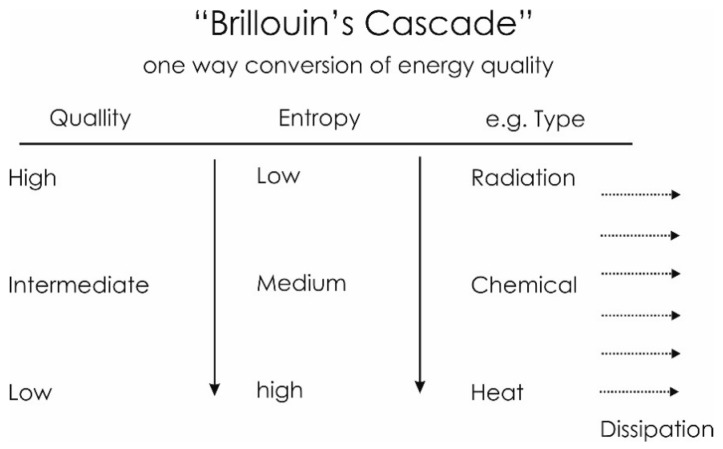
Transformation of energy through “Brillouin’s cascade”. Energy is always transformed in one direction only, from high quality, like radiation, to a sequentially lower quality, ending up as its lowest quality form, namely heat, i.e., ending up as dissipated energy as result of the irreversibility of processes.

**Figure 8 entropy-22-00820-f008:**
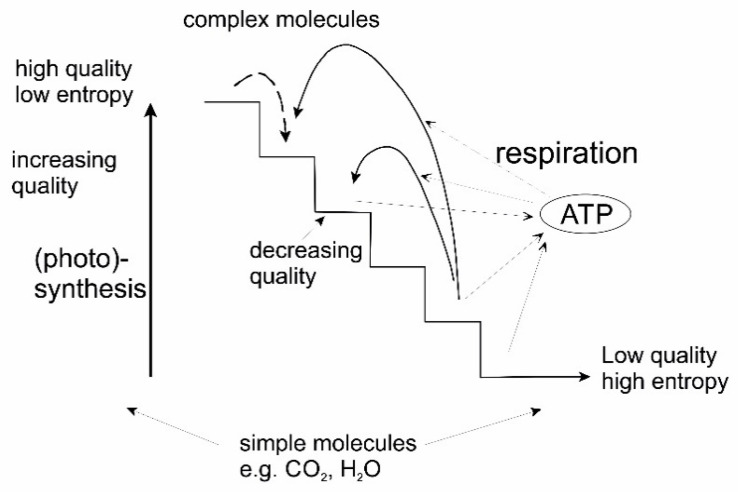
Biological systems move away from thermodynamic equilibrium either (1) as is the case with autotrophs by photosynthesis, i.e., input of high quality energy through solar radiation, or (2) by uptake or build-up of complex molecules which is possible by adding up several energy bundles of intermediate quality via energy carriers, e.g., ATP. In the ecosystem this process is ultimately driven by a supply of chemical energy from the autotrophic organisms, most other processes are driven by this chemical energy, which supplies metabolic and respiratory processes (redrawn and modified from Müller and Nielsen, [178].

**Figure 9 entropy-22-00820-f009:**
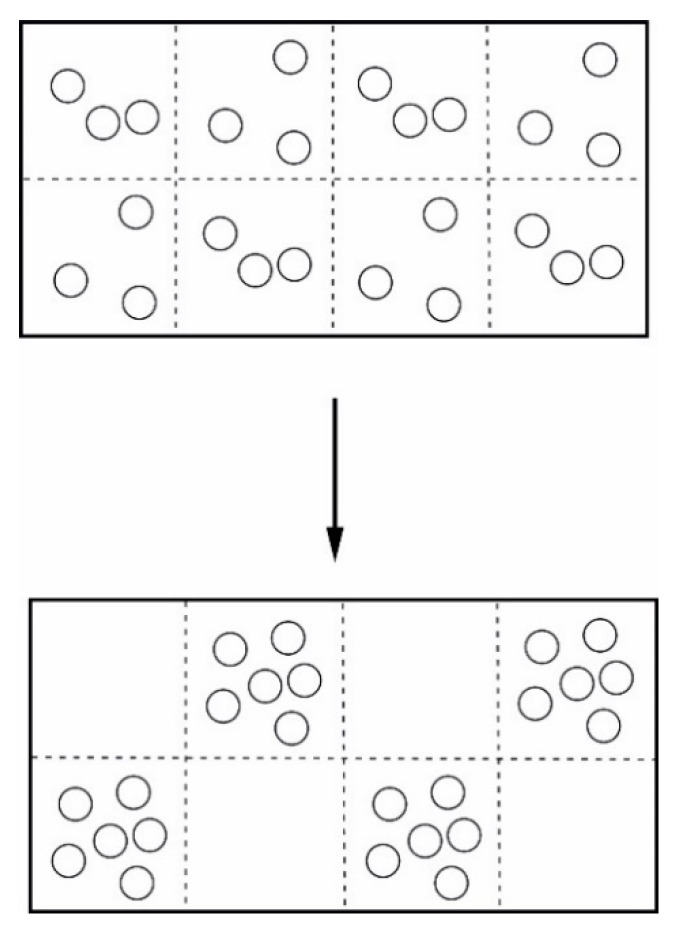
Open systems use the energy flowing through them to create structures which deviate from thermodynamic equilibrium in more and more complex manners. The sorting of elements among the molecules in cells and organisms may be seen as an example of this function of life. (Figure oversimplified and not random enough).

**Figure 10 entropy-22-00820-f010:**
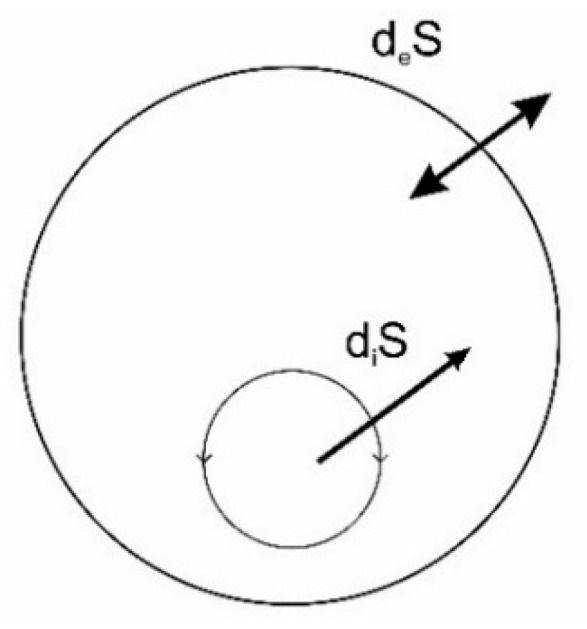
According to Prigogine and co-workers *far-from-equilibrium systems* may be understood also as *dissipative structures* where the total entropy change, *dS*, is a consequence of internal entropy production, *d_i_S*, as well as exchanges with the surroundings, *d_e_S*. Drawing modified from Prigogine [75].

**Figure 11 entropy-22-00820-f011:**
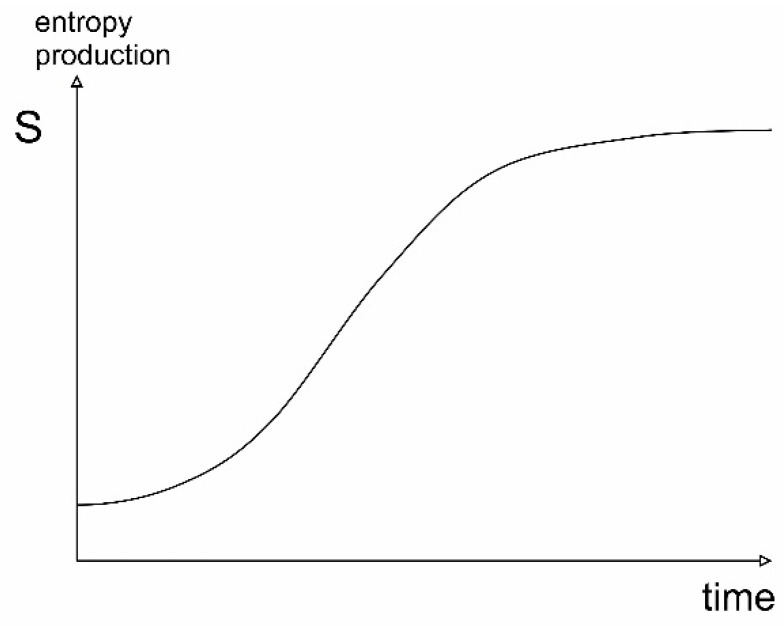
The figure shows the entropy production as a function of time for a system under development. As the system reaches a dynamic equilibrium it enters a state of minimum dissipation and entropy production levels off (redrawn and modified after Prigogine [73].

**Figure 12 entropy-22-00820-f012:**
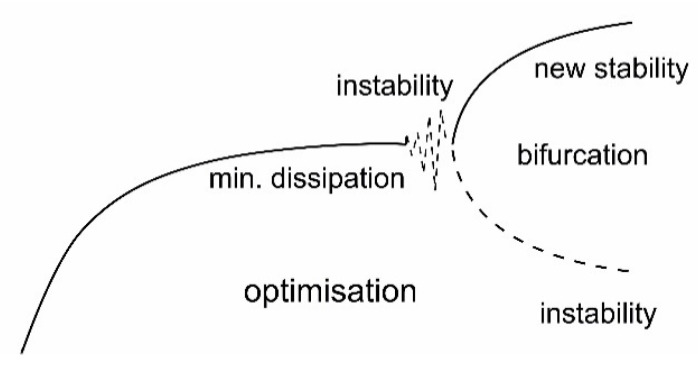
Stable periods with minimum dissipation will eventually lead to instabilities and the possibility for new stable structures to occur. More stable states may coexist depending on the availability of resources and competition for the same, i.e., the *external* as well as *internal constraints* on the system.

**Figure 13 entropy-22-00820-f013:**
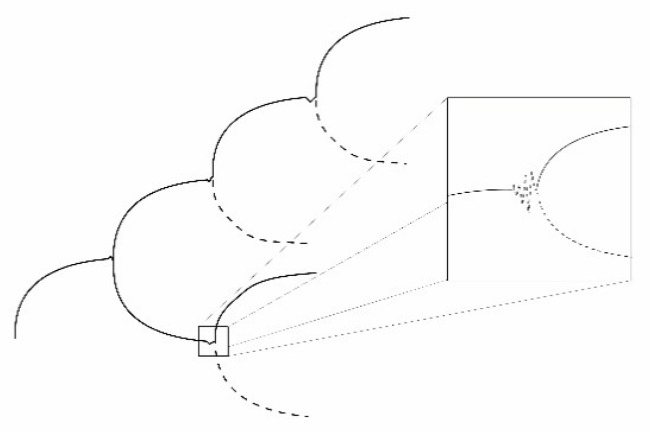
A sequence of minimum dissipation periods, instabilities and bifurcations may be seen as an explanation of serial evolution of biological systems—a so-called habit that is taken on at more levels of hierarchy.

**Figure 14 entropy-22-00820-f014:**
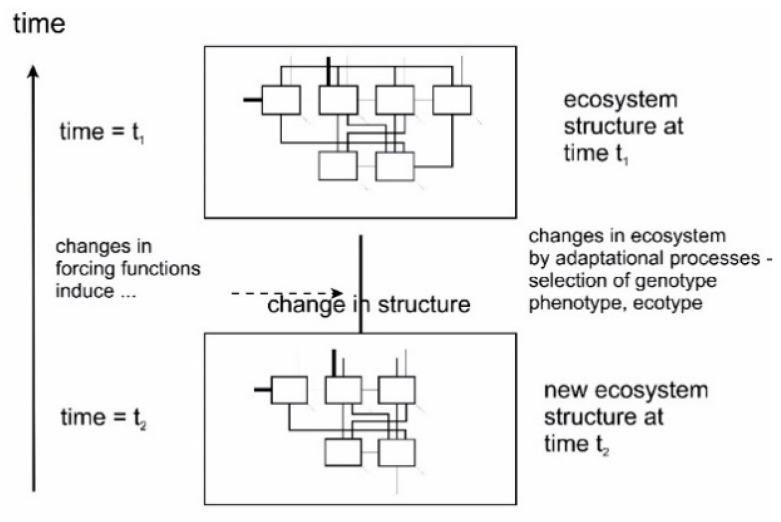
Ecosystems, during their development, show distinct changes in species composition and as a consequence sometimes the whole structure of the trophic network gets affected. This is traditionally seen as a response to changes in factors affecting and driving the system (often in ecological modelling referred to as forcing or control functions) but may eventually also depend on the intrinsic properties of possible organisms. As a consequence, and in order to meet the changes, the ecosystems may react on several levels of hierarchy, e.g., genotype, phenotype or ecotype.

**Figure 15 entropy-22-00820-f015:**
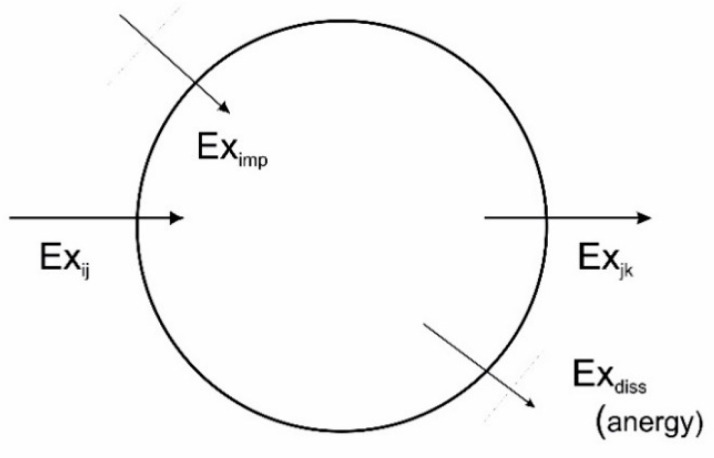
The exergy balance of the system may serve to make the role of various types of energy use more explicit. Exergy may enter over the boundaries, Ex_imp_, like photosynthesis or import of material via forcing functions. Exergy may enter or leave a subsystem from or to other systems parts, Ex_ij_ and Ex_jk_, respectively. Part of the exergy will always be dissipated and lost and will not be available to any system, Ex_diss_ (actually not exergy any longer, but kept open for accounting).

**Figure 16 entropy-22-00820-f016:**
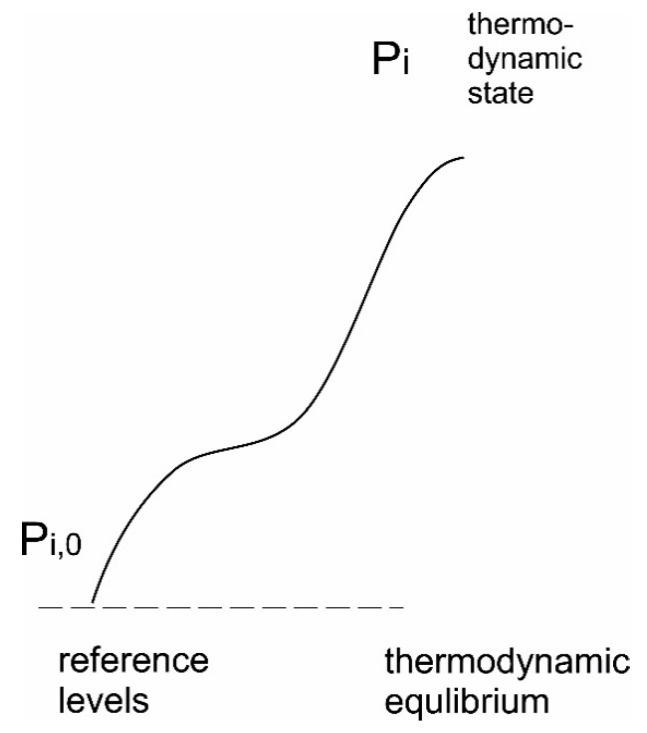
The thermodynamic state, *S* (*p_i_*), of a living system is moved away from thermodynamic equilibrium to a far from equilibrium state. In calculation of exergy the reference level is often set to that of thermodynamic equilibrium *S_eq_* or *S_pi_*_,0_.

**Figure 17 entropy-22-00820-f017:**
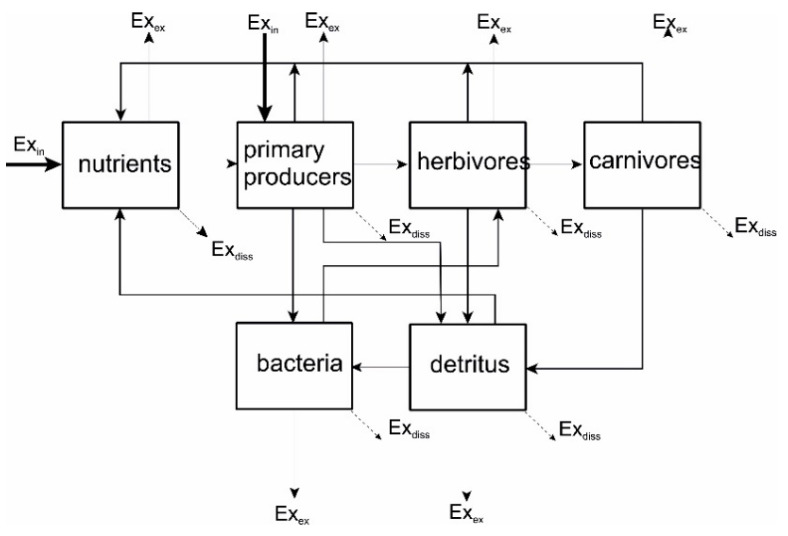
A trophic network illustrating a typical aquatic ecosystem with internal recycling through a detritus and bacterial compartment. The flows are translated into exergy relationships, see Figure 15, making it possible to track the energy conversion and dissipation throughout the system. (redrawn from Nielsen and Ulanowicz [232].

**Figure 18 entropy-22-00820-f018:**
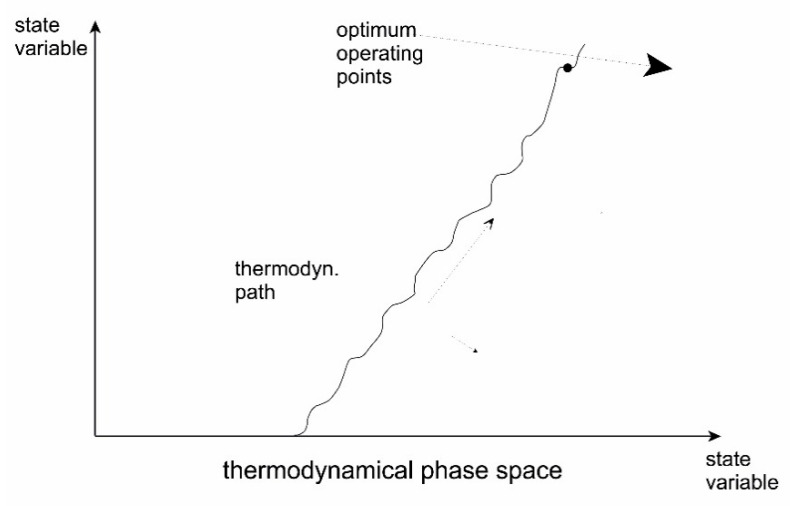
The ecosystem as it evolves will move along a thermodynamic path composed of a sequence of possible thermodynamic (optimal) solutions to the condition met from the surroundings (redrawn and modified from Kay [15].

**Figure 19 entropy-22-00820-f019:**
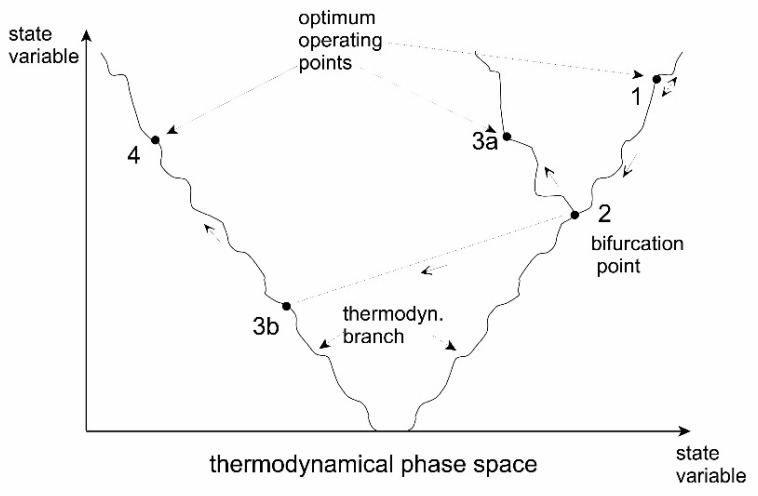
The ecosystem tends to evolve towards an optimum operating point. When subsided to small perturbations the system will stay at or close to its optimum. Major disturbances will move the system to a new optimum operating point in accordance with changes induced. The new path may be found via bifurcations or on other branches through catastrophic events (redrawn and modified from Kay and Schneider [15].

**Figure 20 entropy-22-00820-f020:**
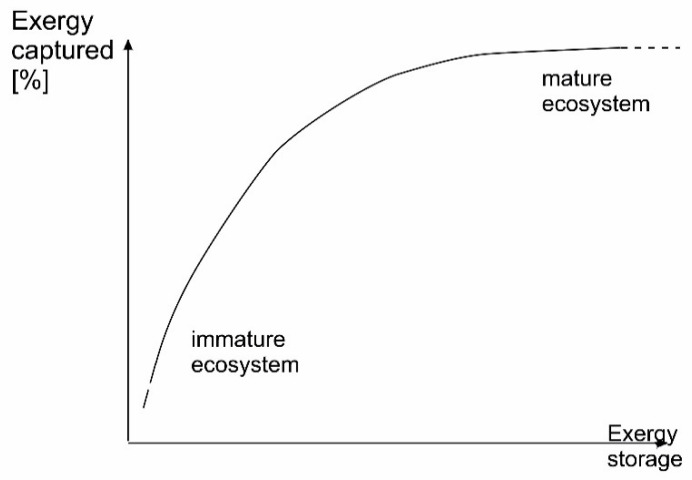
As ecosystems grow and develop, here interpreted as an increase in exergy storage, their efficiency, expressed as percentage of incoming exergy captured, levels off to a seemingly constant level (redrawn and modified from Jørgensen, [7]. Meanwhile, the systems may differ in other parameters such as bio-diversity.

**Figure 21 entropy-22-00820-f021:**
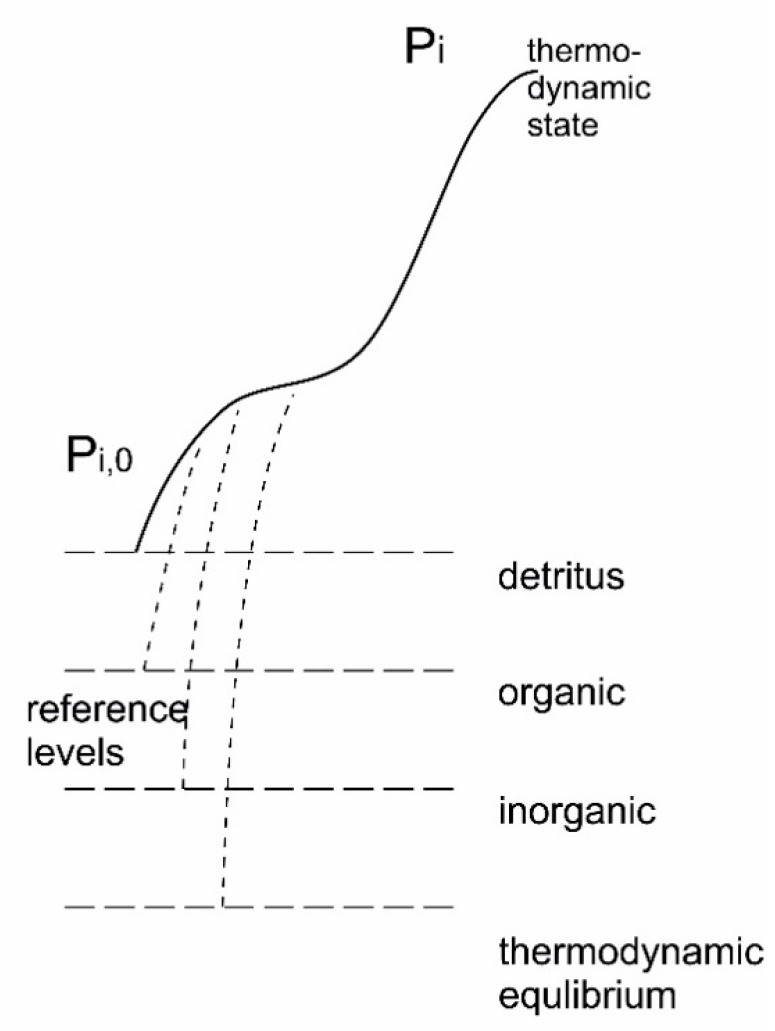
In the calculation of exergy for a biological systems on may choose between different sensible reference levels, detritus level, organic or inorganic level, often referred to as the “Oparinian Ocean” or primordial soup, etc. The exergy contribution from choosing between the various levels is considered to play only a minor part in the calculation of the total exergy of the state. The difference between various reference levels is exaggerated for clarity but is really being considered to contribute only little in the calculation of the total (eco-)exergy state of the system under consideration.

**Table 1 entropy-22-00820-t001:** Examples of thermodynamic properties of ecosystem hypothesized to perform with a pattern-like change during ecosystem development. Some of the properties may be used as indicators of ecosystem state or as candidates of goal functions for instance in ecological modelling.

Variant	Origin (Major References)	Remarks
phenomenology of 24 principles during undisturbed development of naturals systems towards climax society	Odum, E.P. [1,2]	Principle 23 and 24 are referring to decrease in entropy and increase in information of the ecosystem, respectively
emergent properties	Odum, E.P. [21]	The study of emergent properties of ecosystems is proposed as research strategy
maximum (useful) power	Odum, H.T. [3,22,23,24]	The idea originating in Lotka’s papers from the early 1920′ies
eMergy	Odum, H.T. [25,26]	
minimum dissipation/entropy	Mauersberger, P. [27,28,29,30]	minimum dissipation as extremal principle for aquatic ecosystems
entropy	Aoki, I. [31,32,33,34,35,36]	
maximum exergy (storage)	Jørgensen, S.E. [7,37,38]	the exergy function derived was shown to relate to buffer capacity and proposed as a holistic indicator and goal function— exergy optimization of ecosystems recently proposed as an ecological law of thermodynamics
maximum exergy degradation	Schneider, E & Kay, J.J. [14,15,39,40,41]	maximum exergy degradation proposed as driving mechanism, exergy degradation as indicator of ecosystem integrity
maximum entropy production	Martyushev [42,43]	validity of maximum entropy production from physics to biology
Ascendency	Ulanowicz, R.E. [5,6,44,45]	ecosystems as they grow and develop show an increase in ascendency, flows serve as orientor and “stress” indicator
Utility and indirect effect	Patten, B.C. [4,46,47]	Ecosystems flows serve to increase quantitative and qualitative utility of the system Indirect flows are dominating over direct effects by several orders of magnitude
Biomass (maximum)	Straskraba, M. [48] Margalef, R [49]	Biomass as goal function Endosomatic and exosomatic causes

**Table 2 entropy-22-00820-t002:** Historical events important to the development of thermodynamics showing the evolution from the first discoveries implicitly leading to the formulation of the first and second law, and up to our time where the connection to biology was laid out by the establishment of far from equilibrium thermodynamics.

Year(s)	Event	Ref/Source
1789–1791	Lavoiser and Sequin discovers food combustion leading to formation of CO_2_ and H_2_O with a parallel release of heat	after Morowitz [118]
1824	One of the earliest works of Sadi Carnot Betrachtungen über die Bewegende Kraft des Feuers, appears	Carnot 1824 [119]
1865	Clausius’ formulation of the first and second law	Clausius 1865 [120]
1872	Boltzmann search for the so-called H-theorem leading to Boltzmann’s formula	Boltzmann 1872 [121]
1878	Gibbs’ extension of the Boltzmann equation	Gibbs 1878 [122]
1944	Schrödinger states that living organisms are feeding on negentropy and formulates his order form order and order from disorder principles	Schrödinger 1944 [123]
1946	Establishment of far from equilibrium thermodynamics by Prigogine and co-workers (1) understanding of systems as dissipative structures (2) formulation of the minimum dissipation principle (3) evolution through instabilities and bifurcations	Prigogine, 1947 [70] Prigogine and Wiame, 1946 [29] Prigogine and Nicolis, 1971 [124] Prigogine and Stengers [77] Glansdorff and Prigogine, 1971 [125] Nicolis and Prigogine, 1977 [74]
1867	Maxwell’s demon violating the second law	Leff and Rex, 1990 [126]
1967	Brillouin, closer connection to information theory	Brillouin 1960 [127]

**Table 3 entropy-22-00820-t003:** Important milestones in the application of thermodynamic principles to biological and ecological systems.

Year(s)	Event	Main Ref/Source
1922, 1925	Lotka proposes that living organisms compete for energy	after Morowitz [118]
1944	Schrödinger’s states that living organisms are feeding on negentropy and formulates his order form order and order from disorder principles	Schrödinger [123]
1976	Exergy proposed as important factor	Jørgensen and Mejer, 1981 [38] Mejer and Jørgensen, 1979 [37]
1979	Exergy relates to buffer capacity	Jørgensen and Mejer, 1981 [38]
1984	Exergy degradation	Kay, 1984, 1991 [173,222] Kay and Schneider, 1992, [223] Schneider, 1988, [224] Schneider and Kay, 1994a,b,c [15,39,225,226]
1987, 1989	Entropy analysis of lake ecosystems	Aoki [35,227]
1990–1992	changes in ecosystems are generally accompanied by increases in exergy (storage)	Nielsen, 1992 [190] Jørgensen, 1992 [228]
1992	exergy storage used as goal function	Jørgensen, 1992, 1997 [229] Nielsen, 1992 [230]
	exergy relates to: intermediate disturbance hypothesis chaos ascendency the exergy “cushion”	Jørgensen and Padisak, [231] Jørgensen, [228] Nielsen and Ulanowicz [232] Reynolds [233]
1995	New exergy index and specific exergy proposed based on (1) informational content of genome and (2) reference at detritus level	Jørgensen et al. [234] Bendoricchio and Jørgensen [235]
1997	Specific exergy covers other perspectives than the other exergy	Marques et al. [236,237] Xu [238,239]
1997	emergy/exergy ratios	Bastianoni and Marchettini, [58]

**Table 4 entropy-22-00820-t004:** Showing various types of hierarchies organised according to increasing exergy. Simple components are put together constituting more and more complex structures: (a) a cathedral is complex construction eventually composed of bricks of clay, (b) organic molecules are forming cells, forming organs, forming organisms, which eventually are put together in the ecosystems, and eventually constituting the biosphere, (c) likewise the ecological food chain may be viewed as a hierarchy of increasingly complex organisms representing a higher and higher level of exergy.

(a) Architecture	(b) Biological	(c) Ecological
castle, cathedral	biosphere	top carnivore
manor, mansion	ecosystem	carnivore
house	societies	herbivore
stable of bricks	populations	primary producers
pile of bricks	organisms	bacteria
bricks	organs	nutrients
clay	cells	
molecules	cell organelles	
	proteins, enzymes	
	amino acids	
	organic molecules	
	inorganic molecules	
	atoms	

**Table 5 entropy-22-00820-t005:** Weighting factors for various types of organisms based on the information content of the genome expressed during the life cycle of the organism. Weighting is relative to the energetic content of detritus (values taken from [234,235]).

Organism	Number of Information Genes	Weighting Factor
Detritus	0	1
Minimal Cell	470	2.3
Bacteria	600	2.7
Algae	850	3.3
Yeast	2000	6
Fungi	3000	10
Sponges	9000	26
Plants, trees	10,000–30,000	30–90
Worms	10,000–100,000	30–300
Insects	10,000–15,000	30–45
Zooplankton	10,000–50,000	30–150
Crustaceans	100,000	300
Fish	100,000–120,000	300–350
Birds	120,000	350
Amphibians	120,000	350
Reptiles	120,000	350
Mammals	140,000	400
Humans	250,000	700

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
