# Peer review of "Thermodynamics in Ecology—An Introductory Review"

_entropy, 2020, doi:10.3390/e22080820_

Round 1

Reviewer 1 Report

Review of Nielsen et al., Thermodynamics in Ecology – An Introductory Review. Submitted to Entropy (entropy-848511).

The authors present a review of thermodynamic principles as applied to ecology, which is a topic underappreciated by many ecologists. The review is comprehensive, well written, and the authors are eminently qualified to review this field. The introduction to thermodynamics was clear and covered the main points well. The latter sections (Section 7 onwards) were perhaps less comprehensive and focused more on the authors’ own works, but this is to be expected given these authors have led much of the recent work in this field. My comments are few and minor.

Comments:

  1. The readability could be improved through more comprehensive editing and streamlining of the text. The review is quite long, and sections 2–6 are repetitive in parts. Some of the figures and tables also seem unnecessary (e.g. Table 1, Figs 10, 11).
  2. The review lacks a clear development of future steps. To be clear, the review meets its goal of summarising the state of thermodynamics in ecology. What I wondered is whether the authors could give more insight into how ecologists might begin to apply these ideas? One section that caught my eye was the text on goal functions (ll. 1006-1018), which might be a useful way to apply and test thermodynamic principles in ecological systems. I don’t believe these methods are especially computationally challenging given the current levels of quantitative and computational training for most ecology graduates.
  3. The authors repeatedly claim that many of the quantities of interest are unmeasurable, which raises the question of which quantities are indirectly measurable? And how could these be incorporated into empirical ecological studies? As the authors note, there is no single, correct way to study thermodynamics in ecological systems. I personally prefer an approach that is more relaxed about definitions and focused on what we can measure in real ecological systems and would perhaps like to see a bit more consideration of this in the review.
  4. The authors repeatedly mention the relevance of this topic to managers and politicians. Although it is true that applications of ecology require good predictive models, understanding the thermodynamic basis of ecology gets at the fundamental origins of ecological systems. This is potentially ground-breaking and goes well beyond the use of predictive models by managers and politicians.
  5. The latter sections of the review discuss the distinction between exergy storage and dissipation. I believe the authors downplay the potential for these two approaches to be complementary (alluded to in ll. 1342-1352). It may be that greater exergy storage supports greater exergy dissipation (e.g. a more complex ecosystem converting more sunlight to heat than a simpler ecosystem). I’m not claiming that this is the case, I simply believe it is a viewpoint worth considering.

Minor points:

  1. Table 1 is not cited anywhere near where it appears in the introduction, as far as I can tell.
  2. 676-679 mentions an example of a bear “presented earlier” in the text. I can’t find this example.

Author Response

attached in file

Reviewer 2 Report

The authors provide a (mainly historical) review of the use of thermodynamics concepts in ecology.
Publication of this review in Entropy will have the merit to expose broader scientific communities
to the history and state-of-the-art of applying thermodynamics to ecologies.
I therefore recommend publication of this review in Entropy, although the authors should consider the
following recommendations in order to further improve their manuscript:

1.
a. Table 1 appears on page 3, but there is no reference to the table in the text, at least not in
reasonable vicinity to the table. There is a reference
to Table 1 on page 23 line 733, but it seems that in that line the authors try to refer to Table 3.
Table 1 is also references on page 20 (line 1935), but this again seems to be a mistake.

b. I do not understand the table caption "Properties of ecosystem hypothesized perform with a
pattern-like change during ecosystem development." In my opinion this does not reflect the content
of the table.

c. typo in the table: "shown to related to"

2. line 131: Lotka's paper was not published at "the beginning of this century", but at the beginning
of last century.

3. line 171: "the many expressions used does not" -> "do not"

4. lines 217 and 218: something seems to be missing: " -7.5" and " -7.6"

5. Section 2 is a mess, filled with missing / wrong dates ("18??" in table 2, 1924 should be 1824 in line 233,
the reference for work done in 1967 is a paper published in 1960 etc.). The authors should carefully revisit
that section and check the dates.

6. line 261: "activity and dissipation always at the same time results"

7. line 263: "[139], ,"

8. line 303: "Golem ,[175]"

9. I do not understand the sentence on lines 340-342

10. line 351: a ")" too much

11. line 475: This sentence awkwardly ends in "into the closed." Something seems to be missing here.

12. line 503: "energy and heat seem are"

13. line 535: "would therefore should be able"

14. line 536: "was of logically termed"

15. line 598: The last sentence of the caption of figure 11 is incomplete.

16. line 610: I think that "Fig. 8" should be "Fig. 12"

17. There does not seem to be a reference to Fig. 13 in the text.

18. line 644: a superfluous ")"

19. line 705: awkward formulation: "have for some years been practiced world over"

20. line 726: "Seq"

21. In table 3, why are there '??' in "emergy/exergy ratios??"

22. line 737: I so not understand why eq. (19) is called "exergy of the internal energy"

23. line 742: "et al.,[236]"

24. line 768: I do not think that "important milestones" describes the content of table 4

25. line 823: "Size may also play and important role"

26. "from at the beginning"

27. line 860: I do not understand the word "metaphysic" in this context.

28. Eq. (21) and (22) are in contradiction to equations (19) and (20). Equations (21) and (22) do
not refer to an equilibrium / reference state and can therefore not be correct expressions for exergy.

29. line 900: "to be to equal"

30. line 915: the statement "they were hard for many people and in particular scientists to accept" is
strange and is not supported by a reference

31. lines 986-987: The statement "Only the normalized exergy seems to deviate in its behaviour from the others when
compared" is very confusing. A "normalized" quantity is a quantity that differs from the un-normalized one
by a numerical pre-factor (for example, a histogram may be normalized to one to form a probability distribution).
But this numerical (constant!) pre-factor will not change the "behaviour" of the quantity. Am I missing something here?

32. line 1024: "maintaining h the initial set"

33. line 1047: "for extensive overview see. Hihashi ..."

34. line 1169: "for e review of the area"

35. Section 8. There are some major issues with the characterization of thermodynamics and far-from-equilibrium extensions
in the physical sciences. First, thermodynamics is not a "relatively new discipline" (see, for example, line 1220) as it is
more than 150 years old. Second, the most recent references related to far-from-equilibrium generalizations in the
physical sciences are from around 1980. The field of non-equilibrium statistical mechanics has made major progress in
the last 40 years. I of course do not expect the authors to review 40 years of research, but they should be aware that
our understanding of physics far from equilibrium is really very different to what the authors seem to be aware of.
I recommend that they are much more cautious in their discussion and do not make unfounded negatory statements
that are based on a very incomplete knowledge of a very active field of physics.

36. In general, the text is often very verbose and a more succinct discussion is advisable. The authors should also
carefully check the text for typos etc.

Author Response

atached

Reviewer 3 Report

As a theoretical physicist working in the field of non-equilibrium statistical mechanics, including applications to stochastic ecological models, I was highly intrigued by this nicely written review. Its topics certainly fit the broad, interdisciplinary goals of this journal perfectly. Unfortunately, though, I feel that the presentation at least should be improved: there are several errors and misconceptions that please need to be corrected or eliminated; various conceptual problems, some quite severe, should be mentioned in the main text already, and not entirely deferred to the concluding two sections; and at several instances, it would be helpful if some more details could be included.

I would urge the authors to please carefully address the following items, and amend their manuscript accordingly.

  • It is inappropriate to state that thermodynamics pertains merely to classical ideal gases. That was never a true statement, never mind inadequate expositions in some undergraduate textbooks. By the way, in physics we would not call thermodynamics a "young discipline", as in fact it is one of the oldest in our field. Even statistical mechanics is now 150 years old.
  • The authors need to please be very careful when talking about physical quantities and their changes under some dynamics. The second law addresses entropy *changes* (not that the entropy itself needs to be positive, as wrongly stated on pages 5, 9, ...).
  • While it is frequently stated that entropy could be "explained" through disorder (page 8), that is just in general a misleading statement too and should please be avoided.
  • I was distressed to see the utterly wrong thermodynamic statement of the second law in eq. (3), page 10. Only the reversible (quasi-static) heat exchange is related to entropy change, del Q_rev = T dS. The second law states that under *irreversible* conditions, dS_irr > 0, del Q_irr < T dS as there are inevitable dissipative heat "losses" to the environment. (The authors' inequality would imply that del Q > 0 always, which is obviously nonsense.)
  • Also quite unfortunately, Figure 3 (and also 9) is badly misleading; in the lower row, the figures actually show quite special and hence rare configurations, with each small box filled with an equal number of particles. That is certainly far from the most likely configuration with highest entropy. And no, entropy in general has nothing to do with "a spatial arrangement of the parts of a system" (page 11).
  • Eq. (6) on page 11 is just the first law of thermodynamics, it does not contain the second law at all; and another misconception: it is perfectly standard fare in thermodynamics to add the chemical potential / work contributions as in Eq. (7).
  • There are some more subtle errors. In truly fully isolated systems, without any interactions with the environment, the entropy never changes. The second law pertains to systems that are in (weak) contact with the external world, and for which some constraints are released.
  • The authors frequently refer to certainly seminal, but by now rather dated work by Prigogine and collaborators, unfortunately ignoring the considerable amount of progress in non-equilibrium statistical physics achieved over the past at least three decades. While I do not expect this shortcoming to be truly amended in a revision, I would still like to point out that a number of historic assertions have been shown to be wrong: No, non-equilibrium steady states are definitely not in general governed by minimal dissipation (only some are); so this certainly cannot serve as an overarching principle (page 6 and elsewhere). More broadly, there are known shortcomings in the attempts to construct a "steady-state thermodynamics"; see, e.g., R. Dickman and R. Motai, Phys. Rev. E 89, 032134 (2014).
  • I find the text at many instances too vague in defining its concepts and fundamental properties; examples: what is meant by "optimizing thermodynamic efficiency" (page 6), what are the "qualities" of energy precisely (Brillouin's classification truly is badly out of date), what is meant with "optimal operating point" (page 32) - optimal relative to what quantity? - etc.
  • The discussion of dissipative structures could be made clearer. I would have stated 0 < dS_tot = dS + dS_env, whence dS < 0 for the (e.g., living) system in question merely necessitates for the environment dS_env > |dS| > 0 (page 16) rather than referring to murky "internal" and "external" processes.
  • I really don't see how dividing by the total mass changes anything fundamental in eq. (15), page 17; and again, as pointed out above, stationarity merely requires dS / dt = 0, *not* minimal dissipation (page 18). There certainly *can* be instabilities for the system - which should be more aptly referred to as "phase transitions" rather than bifurcations, as this pertains to systems with multiple degrees of freedom - but that is not a necessity. I also note that of course the "rigidity" of a (metastable) stationary state against external perturbations (page 32) is a tautology (hence trivial). It is however wrong to assert that increasing applied forcings will necessarily increase rigidity. There is no general basis for this statement.
  • For the application of purely thermodynamic concepts involving energy, exergy, etc., I have two fundamental conceptual objections: (1) Just the energy really is only relevant for equilibrium or near-equilibrium systems. Once a system is taken far from thermal equilibrium, it is not always a relevant concept. For example in biology, evolution is certainly constrained by the laws of thermodynamics, sure - but also driven by reproduction and the ensuing prevalence of genes that are "effective" in this process, which goes far beyond energetics. (2) Temperature is purely a thermal *equilibrium* concept; while it can be generalized to near-equilibrium systems in the linear response regime or when local thermal equilibrium is a decent approximation, it loses its meaning entirely in far-from-equilibrium systems. Hence I really do not understand what the variable "T" should refer to in an ecological system, Eqs. (19), (20) on page 23 ff.
  • Expressions such as (23) and (24) that are linear in the concentrations and their logarithms are likely valid only in a dilute limit, correct? But that does not seem to be applicable under generic ecological conditions. I note that perhaps the exergy concept here can be related to what statistical physicists refer to as "mutual entropy".
  • Despite these shortcomings, some of which are discussed to some extent in the article's final sections, I consider the applications listed in Section 7 highly intriguing, but found the explanations and discussions too terse to be truly useful. I would definitely welcome some more expanded expositions.
  • Finally, please eliminate (somewhat annoying...) typos in authors' names: Boltzmann, Gibbs, ... ; on page 7, Carnot's work of course dates to 1824 (not 1924).
